# Unlocking Zero-Shot Geospatial Reasoning via Indirect Rewards

**Chenhui Xu** [1] [*]  **Fuxun Yu** [2]  **Michael J. Bianco** [2]  **Jacob Kovarskiy** [2]  **Raphael Tang** [2]  **Qi Zhang** [2]  **Zirui Xu** [2]
**Will LeVine** [2]  **Brandon Dubbs** [2]  **Heming Liao** [2]  **Cassandra Burgess** [2]  **Suvam Bag** [2]  **Jay Patravali** [2]
**Rupanjali Kukal** [2]  **Mikael Figuero** [2]  **Rishi Madhok** [2]  **Nikolaos Karianakis** [2]  **Jinjun Xiong** [1]

## Abstract

Training robust reasoning vision-language models (VLMs) in rare domains (such as geospatial) is fundamentally constrained by supervision scarcity. While raw geospatial imagery is abundant, the amount of task-**direct** supervision falls far behind that of common domains. In this work, we validate an important conclusion: **indirect** verifiable rewards, derived from seemingly unrelated metadata, are sufficient to induce sophisticated and generalizable geospatial reasoning across a wide range of downstream tasks (25+). We present Geo-R1 as one empirical instantiation of this paradigm. Rather than relying on limited task-specific annotations (i.e., direct rewards), Geo-R1 utilizes scalable, verifiable **indirect** proxy rewards based on cross-view alignment with metadata (geolocation information) to drive reinforcement learning at scale. Such indirect rewards successfully motivate the model to discover and internalize zero-shot geospatial reasoning across diverse tasks, achieving extraordinary zero-shot transfer on out-of-distribution benchmarks and even surpassing fully supervised specialists on certain benchmarks. These findings indicate that optimizing for **indirect verifiable rewards** may provide a scalable pathway to unlock generalized reasoning capabilities in rare domains with massive unlabeled data archives. Our code is availavle at: https://github.com/miniHuiHui/Geo-R1.

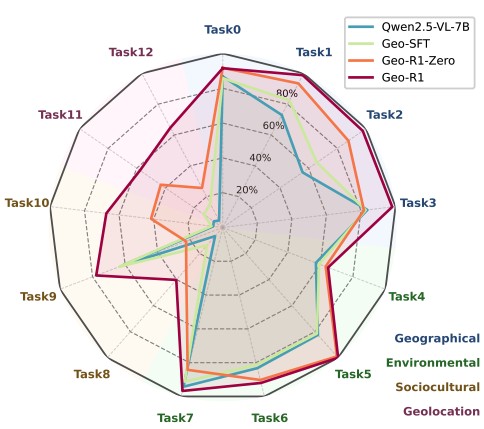

*Figure 1.* Geo-R1 significantly outperforms baseline (Bai et al., 2025) across 13 verifiable geo-reasoning tasks on the GeoChain benchmark (Yerramilli et al., 2025) in the zero-shot setting. See Table 9 for detailed description of these tasks.

## 1. Introduction

Training vision-language models (VLMs) with reasoning capabilities for rare domains such as geospatial faces a fundamental data asymmetry: the geospatial field possesses an inexhaustible archive of imagery (satellite, drone, ground-view, etc.) but suffers from non-scalable task-**direct** supervision (Mall et al., 2024). As shown in Fig. 2, most scalable supervision accompanying these raw geospatial images are those sparse metadata, such as GPS coordinates and timestamps, which usually seems unrevelant since they lack the dense, rich reasoning semantics required for traditional fully supervised (SFT) or weakly supervised fine-tuning. Such non-scalable direct supervision often causes SFT-based models to learn only narrow task distributions, resulting in brittle performance and out-of-distribution (OOD) generalization.

Reinforcement learning (RL) potentially offers an escape, as evidenced by the success of reasoning models in math and code (Guo et al., 2025). However, the same roadblock persists: the limited scale of high-fidelity, verifiable **direct** rewards for the wide range of downstream tasks (classification, detection, segmentation, captioning, QA, REC/RES, open-ended reasoning, etc.). Metadata such as geolocation provides a scalable source of labels paired with every image, but the challenge is that these signals do not explicitly reward the complex visual reasoning required to understand

---

[*] Work was done during internship at Microsoft. [1]University at Buffalo [2]Microsoft. Correspondence to: Chenhui Xu <cxu26@buffalo.edu>, Jinjun Xiong <jinjun@buffalo.edu>, Fuxun Yu <fuxunyu@microsoft.com>.

*Proceedings of the $43^{rd}$ International Conference on Machine Learning, Seoul, South Korea. PMLR 306, 2026. Copyright 2026 by the author(s).*

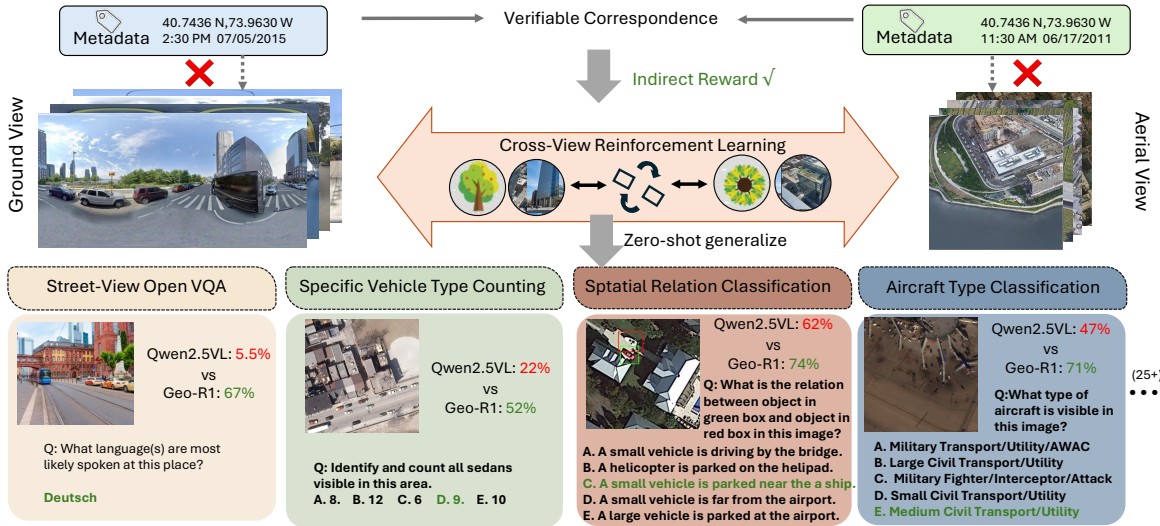

*Figure 2.* **Concept of Indirect Reward.** Although metadata is just indirect supervision signal to geospatial images, its correspondence can be verifiable reward signals for improving VLM's reasoning process towards broader geospatial understanding for different views.

a scene. This leads us to think the pivotal question:

*To what extent can we unlock geospatial intelligence within VLMs using only indirect signals like metadata?*

We draw a key insight from prior cross-view CLIP work (Mall et al., 2024), and design a framework that learns to align ground-level and aerial imagery from the same location as a high-stakes proxy for geospatial reasoning. We find that although location matching is an *indirect* signal providing only a binary reinforcement indicating whether two divergent views share the same coordinates, it can enforce a rigorous understanding of objects and scenes, including recognition, counting, and spatial correspondence. For example, to achieve a match, the model must synthesize fragmented visual cues across divergent scales by implicitly aligning building footprints with facades, inferring orientation from sun angles, etc. Although the indirect reward signal is purely outcome-oriented and provides no explicit guidance on the reasoning process during RL, the model is nevertheless motivated to autonomously conduct complex analysis over both ground- and aerial-view images to bridge "ground-to-air" evidence.

We reveals that such indirect rewards can incentivize the model to internalize a robust, transferable logic of the physical world understanding, thus leading to the emergence of broad **zero-shot** geospatial reasoning capability. We materialize this as Geo-R1, a post-training framework designed for indirect reward-driven RL. Empirical evaluations reveal that by leveraging metadata-based cross-view alignment as scalable rewards, Geo-R1 achieves substantial **zero-shot** improvements across 25+ geospatial tasks, as illustrated in Fig. 1 and Fig. 6, effectively unlocking the latent intelligence within unlabelled archives and establishing a new

frontier for OOD geospatial generalization.

**Contributions.** We make the following contributions:

- We reveal, both empirically and theoretically, that optimizing for a indirect reward compels the vision-language models to internalize view-invariant geospatial understanding rather than superficial shortcuts.

- We bridge the gaps of the fundamental data asymmetry in geospatial intelligence, where the volume of raw imagery exceeds text annotations by orders of magnitude.

- We introduce Geo-R1, a framework harmonizing reasoning scaffolding and RL elevation, which achieves extraordinary zero-shot performance across diverse OOD benchmarks, from both areial- and ground-view.

## 2. Related Works

**Geospatial Foundation Models.** Recent geospatial foundation model training paradigms span from general-purpose visual pretraining, e.g., masked auto-encoding (Cong et al., 2022; Szwarcman et al., 2024; Reed et al., 2023), to contrastive learning (Zhang et al., 2024; Li et al., 2023; Liu et al., 2024b) and remote sensing VLMs (Kuckreja et al., 2024; Muhtar et al., 2024; Pang et al., 2025). While these models excel at specific tasks like representation learning, detection, retrieval, geospatial VQA, most cannot conduct reasoning (e.g., decomposing a task into bearings, distances, landmarks and synthesize information) nor do they refine thinking process to make final decisions.

**Inference Time Scaling with Reinforcement Learning.** Inference-time scaling improves reasoning by allocating more test-time compute via RL-based post-training.

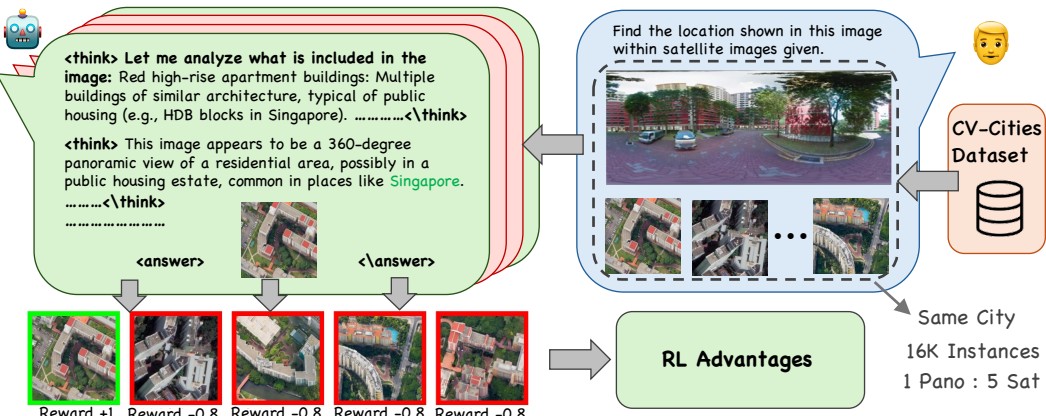

*Figure 3.* Cross-view pairing task for reinforcement learning with verifiable rewards.

Outcome- and process-based rewards, human/AI feedback (RLHF/RLAIF), and verifier-guided RL have been shown to produce longer, more structured chains with higher factuality (Christiano et al., 2017; Bai et al., 2022; Kumar et al., 2025). Recent "R1-style" systems train policies (LLMs/VLMs) to generate and self-verify solutions with group-relative or advantage-normalized objectives to stabilize long-horizon updates (Guo et al., 2025; Shao et al., 2024). These approaches to training models via RL typically require logically straightforward, strong reward signals. Most recently, GLOBE (Li et al., 2025a) explore geolocation task with a direct-reward-based RL, getting some in-distribution performance gains, serving as the pioneer for deploying advanced RL techniques in geospatial task.

Most aforementioned works target natural scene VQA, chart/table reasoning, or math/logic. In this work, we position geospatial reasoning as a primary goal: the model must reason across views, decompose the task into geo-primitives, and justify decisions with intermediate thoughts. To scale up RL in geospatial settings, verifiers should be programmatic and precise, enabling dense, low-latency rewards without human annotation. Our approach integrates inference-time scaling (self-consistency and verifier-guided search) with RL that learns to self-explore intermediate states that improve geo-metrics, closing the loop between perception, reasoning, and measurable correctness.

## 3. Indirect Verifiable Rewards

The core challenge in RLVR lies in designing a reward function that balances scalability with the depth of induced reasoning. While geospatial metadata provides scalable indirect supervision signal, transforming these noisy signals into high-quality reasoning behaviors requires a theoretically grounded formulation. As shown in Fig. 3, We propose a proxy RL task: *Cross-View Pairing*.

**Cross-View Pairing.** Formally, let $I_g$ be the ground-level panorama image and $\mathcal{S} = \{I_s^1, \ldots, I_s^k\}$ be a set of candidate satellite images. This set contains exactly one ground truth $I_s^*$ and $k-1$ hard negatives. The information content of any image $I$ can be decomposed into view-invariant geometric semantics $\Phi(I)$ and we define modality-specific nuisance factors as $N(I)$.

The objective of our RL training is to optimize the policy $\pi_\theta$ to generate a reasoning chain $C$ that maximizes the probability of identifying $I_s^*$. Specifically, we employ a binary $\{-1,1\}$ reward to reward/penalize correct/failure choices. Crucially, by sampling hard confusers from the same spatiotemporal neighborhood as $I_s^*$, we ensure that nuisance factors are statistically non-discriminative within the candidate set: $\mathcal{I}(Y; N(\mathcal{S})) = 0$. We formalize the impact of this design via the following theorem:

**Theorem 3.1** (Invariance Emergence via Hard-Negative Bottleneck). *Under the condition where nuisance factors $N(\mathcal{S})$ are statistically independent of the target identity $Y$ while geometric semantics $\Phi(\cdot)$ uniquely identify $Y$, maximizing the mutual information between the reasoning chain $C$ and the target $Y$ is mathematically equivalent to maximizing the mutual information with the view-invariant features $\Phi(I_s^*)$:*

$$\max_\theta \mathcal{I}(C; Y \mid \mathcal{S}) \iff \max_\theta \mathcal{I}(C; \Phi(I_s^*))$$
$$s.t. \quad \mathcal{I}(C; N(I_s^*)) \to 0 \quad (1)$$

**Learnability and Non-Triviality Analysis.** Theorem 3.1 positions the task in the optimal difficulty regime for RL that it is theoretically solvable yet computationally non-trivial. We prove this using conditional entropy analysis:

1. *Non-Triviality (Lower Bound):* Since nuisance factors are strictly uninformative regarding the target ($\mathcal{I}(Y; N) = 0$), any reasoning policy $\pi_{shortcut}$ relying exclusively on superficial features $N$ yields maximum entropy: $\mathcal{H}(Y \mid \pi_{shortcut}) = \log K$. This prevents trivial solution collapse and forces exploration.

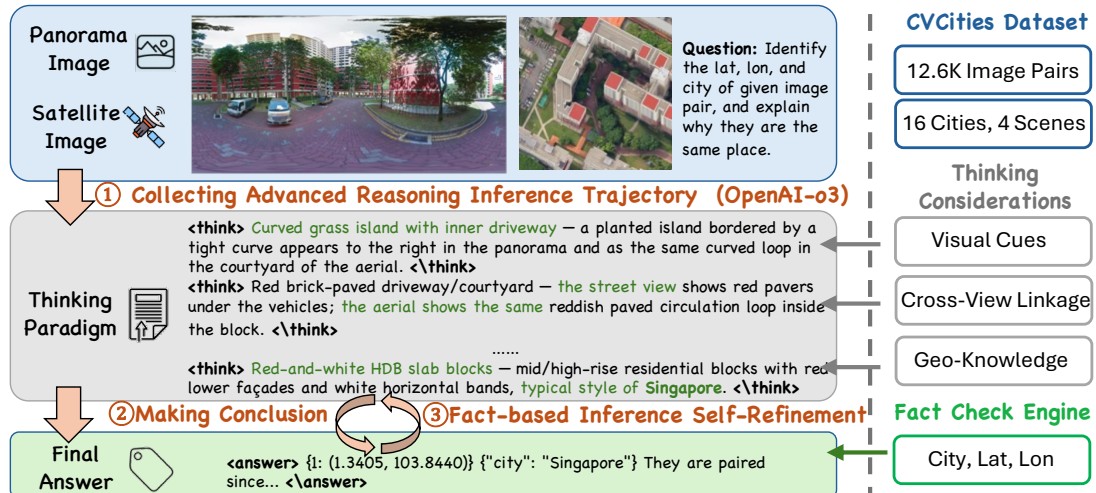

*Figure 4.* Geospatial thinking CoT data engine.

2. *Solvability (Upper Bound):* Despite the difficulty, the task remains solvable because the physical world is consistent. The intersection of ground and satellite semantics is non-empty: $\Phi(I_g) \cap \Phi(I_s^*) \neq \emptyset$. Thus, there exists an optimal policy $\pi^*$ that extracts $\Phi$, reducing entropy to zero: $\mathcal{H}(Y \mid \pi^*) \to 0$.

This entropy gap $\Delta\mathcal{H} = \log K$ represents the "reasoning margin" that the RL process must bridge, ensuring the model learns robust features rather than fitting noise.

**Mechanism of OOD Generalization.** Finally, we establish the logical bridge between this proxy task and zero-shot generalization. The solving of Cross-View Pairing effectively demands the model to approximate a learnable transformation function $F : \Phi(I_g) \to \Phi(I_s)$, which maps ground-level observations to overhead layouts. Since $\Phi$ consists of geometric laws (e.g., perspective projection, collinearity, object permanence) rather than texture statistics, the function $F$ is *domain-agnostic*. Mathematically, let $\mathcal{D}_{train}$ and $\mathcal{D}_{test}$ be two disjoint distributions (e.g., different cities). Since laws of projection governing $\Phi$ are invariant across locations:

$$P(Y \mid \Phi, \mathcal{D}_{train}) = P(Y \mid \Phi, \mathcal{D}_{test}) \qquad (2)$$

Therefore, by forcing the reasoning chain $C$ to encode $\Phi$ via the hard-negative bottleneck, the model inadvertently learns a universal geospatial parser. This allows the reasoning behaviors to transfer zero-shot or to out-of-distribution scenarios, as the underlying causal logic remains constant even when the visual environment changes.

## 4. Geo-R1: Learning with Indirect Rewards

Geo-R1 is designed to answer a singular question: How can we induce complex geospatial reasoning without expensive expert annotations? Our framework answers this by harmonizing two distinct stages: (1) **Scaffolding**, which

instills a basic "thinking paradigm" via supervised fine-tuning (SFT), and (2) **Self-Exploring**, which unlocks deep self-incentivized reasoning through Reinforcement Learning with Verifiable Rewards (RLVR) using our novel indirect reward signal. We include the whole design details of Geo-R1 framework in Appendix A, B, C.

### 4.1. Stage 1: Geospatial Thinking Scaffolding

Before a model can learn from indirect rewards, it must first understand how to reason about geography. We construct a "thinking scaffold" not to teach specific answers, but to instill a domain-generic reasoning structure.

**The CoT Data Engine.** We synthesize a small, high-quality dataset of 12.6k reasoning trajectories using the CV-Cities dataset. As shown in Fig. 4, instead of diverse QA pairs, we focus on a single paradigm: analyzing visual cues (e.g., vegetation, road markings), corroborating evidence across views, and linking them to geospatial knowledge (e.g., climate bands). We employ a Fact-Check Engine to verify key entities (coordinates, city names) against metadata, ensuring the initial scaffold is grounded in reality.

**Paradigm Injection.** Diverging from the traditional SFT philosophy of "cold-starting" on diverse tasks, we implant a unified geospatial reasoning paradigm derived from a single template. This approach retains the benefits of accelerating RL warm-up and fostering transferable reasoning, yet significantly minimizes SFT task diversity to prevent catastrophic forgetting. Furthermore, this unified synthesis proves highly efficient, circumventing the extensive annotation overheads typical of the data-scarce geospatial domain.

### 4.2. Stage 2: Reasoning Elevation via Indirect Signals

Building on the scaffolded model established in Stage 1, we advance to the Elevating stage by applying the Cross-View

Table 1. Results on in-distribution cross-view pairing task.

| Model | Accuracy | Completion Length |
|-------|----------|-------------------|
| Qwen2.5-VL-7B | 19.0% | 204.6 |
| Geo-SFT | 23.1% | 1127.6 |
| Geo-R1-Zero | 78.1% | 587.4 |
| Geo-R1 | **82.4%** | 378.8 |

Pairing task defined in Sec. 3.

**Reward Formulation.** Crucially, we leverage strictly outcome-based rewards, supervising the model solely on the verifiability of its final prediction rather than imposing human priors on the intermediate reasoning process. We optimize the model using Group Relative Policy Optimization (GRPO) (Shao et al., 2024) with a composite reward:

$$r = \lambda_{\mathrm{acc}}\, r_{\mathrm{acc}} + \lambda_{\mathrm{fmt}}\, r_{\mathrm{fmt}} + \lambda_{\mathrm{len}}\, r_{\mathrm{len}} + \lambda_{\mathrm{rep}}\, r_{\mathrm{rep}}. \quad (3)$$

where $r_{\mathrm{acc}}$ serves as the primary performance signal based on verifiable correctness, while $r_{\mathrm{fmt}}$, $r_{\mathrm{len}}$ and $r_{\mathrm{rep}}$ function as auxiliary regularizers to ensure structural validity, reasoning conciseness, and repetition penalties. We explain the details of rewards in Appendix C.

### 4.3. Settings and Implementation Details

We implement Geo-R1 with LLama-Factory (Zheng et al., 2024) and VLM-R1 (Shen et al., 2025), two open-source LLM post-training frameworks for fast and stable SFT and GRPO training. We use Qwen2.5-VL-7B (Bai et al., 2025) as the base model, and then conclude a Geo-SFT intermediate state model after the stage-1 scaffolding-oriented SFT. Starting from Geo-SFT model, we conduct the stage-2 training and get the final Geo-R1 model. We also conduct RL training independently, starting directly from the base model, resulting in Geo-R1-Zero. We conduct full-parameter fine-tuning on the model for more stable convergence and higher final accuracy. We train the model on 8×NVIDIA H100 GPUs. We employ vLLM (Kwon et al., 2023) to accelerate model inference during RL and testing phases. Training details are provided in the Appendix G.

## 5. Empirical Performance of Indirect Rewards

In this section, we present the experimental results and provide corresponding remarks during training stage and testing on standard benchmarks. We evaluated the model on both in-distribution and out-of-distribution datasets, while the latter encompassing diverse perspectives and multiple tasks per view. Our results demonstrate that RL with a indirect reward fosters robust geospatial reasoning, enabling zero-shot generalization across these unseen tasks.

### 5.1. In-Distribution Geospatial Correspondence

**Benchmark.** We evaluate the in-distribution cross-view pairing task. We sample 5,000 holdout sets of images

Table 2. Comparison with task-specific geo-localization model on IMAGEO-GSS (Li et al., 2025b) dataset.

| Metric | GeoCLIP | Geo-R1 |
|--------|---------|--------|
| City Accuracy↑ | 0.1086 | **0.3272** |
| Country Accuracy↑ | 0.6361 | **0.8146** |
| Mean Distance (km)↓ | 943.4838 | **568.3228** |
| Median Distance (km)↓ | 266.8987 | **69.4008** |

from CV-Cities using the same method as described in appendix C.1, to serve as the test set. They do not overlap with the data points sampled during either the SFT or RL stage.

**Remark 1: Supervised Learning Fails on Capturing Indirect Signals.** We found that learning cross-view pairing using only positive examples through SFT does not generalize well. As shown in Table 1, the Geo-SFT model can only marginally outperform the base model by 4.1%, which is still extremely close to random guess (20% accuracy). We also observe a significant increase in the completion length of the Geo-SFT model attributed to substantial content duplication, which indicates heavy SFT is not a good fit for complex and generalizable geospatial reasoning tasks.

**Remark 2: Indirect Rewards Drive the Discovery of Robust Visual Correspondence.** RL delivers robust performance improvements for the Cross-View Pairing task through both positive and negative instances feedback. As shown in Table 1, both Geo-R1 and Geo-R1-Zero model achieve a significant performance boost in terms of about 60% accuracy gain. This means during the RL process, the model does not merely memorize images but learns how to distinguish images from multiple viewpoints.

**Completion Length.** Regarding completion length, thanks to the length reward and repetition penalty, the inference completion length of Geo-R1 and Geo-R1-Zero is kept within a reasonable range, avoiding the excessive repetition seen in Geo-SFT. Benefiting from the thinking paradigm learned during the scaffolding SFT phase, Geo-R1 exhibits a more concise and regular intermediate reasoning process compared to Geo-R1-Zero, demonstrated by a completion length approximately 1/3 shorter.

### 5.2. Out-of-Distribution Geospatial Reasoning

A pivotal attribute of the indirect-reward paradigm is its capacity to unlock the latent, open-ended geospatial reasoning capabilities of VLMs. To rigorously validate this, we subject Geo-R1 to a comprehensive evaluation on diverse OOD benchmarks. These evaluations span heterogeneous modalities (aerial and ground-view) and complex reasoning tasks, all conducted under strict **zero-shot settings**.

#### 5.2.1. GEOCHAIN: GROUND COMPLEX REASONING

**Benchmark.** GeoChain (Yerramilli et al., 2025) is a geospatial reasoning benchmark which employ template-based chain-of-thought to solve a street-view geolocation task

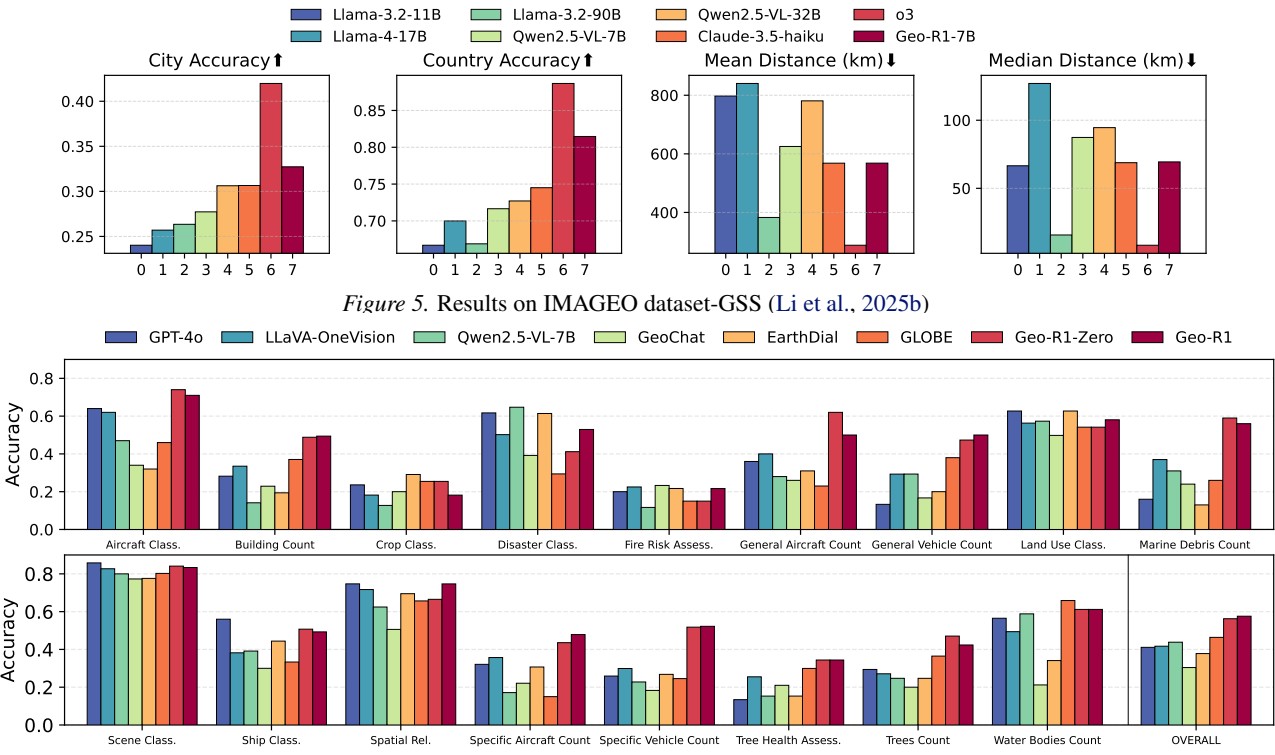

*Figure 5.* Results on IMAGEO dataset-GSS (Li et al., 2025b)

*Figure 6.* Performance comparison on GeoBench-VLM-CLS (Danish et al., 2025)

across 20 cities, of which 15 cities are OOD. We did not use their CoT template but allow free-form reasoning of our models to answer the questions. We evaluated the model's performance on 13 subproblems with explicit ground-truth data. See GeoChain problem details in Appendix I.

**Remark 3: Proxy-Task Optimization Unlocks Zero-Shot Generalization.** Crucially, the reasoning logic learned solely through the proxy task (indirect rewards) transfers zero-shot to complex, unseen geospatial benchmarks like GeoChain. As shown in Fig. 1, the Geo-R1 optimized with indirect rewards significantly outperforms baselines across diverse categories (Cultural, Environmental, Geographical). This proves that the model is not overfitting to the proxy task, but has internalized a universal geospatial parser.

### 5.2.2. IMAGEO-BENCH: STREET-VIEW GEOLOCATION

**Benchmark.** The IMAGEO-Bench (Li et al., 2025b) is a systematic OOD benchmark that evaluates large language models' ability to perform image geolocalization by testing accuracy, distance error across diverse datasets of global (6152 images from 396 cities) and US (2928 images).

**Remark 4: Geo-R1 Outperforms Open-Source LLMs.** As shown in Fig. 5, we evaluated multiple open-source and closed-source models on the IMAGEO-GSS dataset. Our results show that Geo-R1 achieves the highest city and country identification accuracy among all open-source models. Note Llama-3.2-90B (Dubey et al., 2024) appears lower

*Table 3.* Performances on MP16-Reason (Li et al., 2025a).

| Method | Street 1km | City 25km | Region 200km | Country 750km | Continent 2500km |
|---|---|---|---|---|---|
| ISNs (Muller-Budack et al.) | 26.24 | 47.38 | 55.88 | 68.48 | 80.92 |
| GeoCLIP (Vivanco Cepeda et al.) | 29.28 | 52.52 | 66.85 | 84.07 | 93.33 |
| Hybrid (Astruc et al.) | 0.97 | 16.53 | 28.72 | 50.31 | 71.47 |
| RFM-YFCC (Dufour et al.) | 11.72 | 46.64 | 60.46 | 77.97 | 91.96 |
| Qwen2.5-VL-7B (Bai et al.) | 15.42 | 52.72 | 62.86 | 75.11 | 83.47 |
| InternVL3-8B (Zhu et al.) | 12.01 | 44.17 | 55.66 | 75.36 | 86.98 |
| *Task-specific reasoning supervision* | | | | | |
| GeoReasoner-7B (Li et al.) | 10.06 | 40.44 | 50.91 | 68.01 | 79.68 |
| GLOBE-7B (Li et al.) | 17.99 | 62.85 | 73.83 | 86.68 | 92.52 |
| Geo-R1 (Ours) | 17.98 | 61.02 | 73.91 | 86.78 | 93.56 |

mean and median distance since they calculate it on top of successful responses only (success rate of 46%), while ours successful response rate are 99%. The close-source o3 continues to hold an absolute lead in this benchmark, which we attribute to its tremendous parameter scale and reinforcement learning efforts. We include more details, including latitude, longitude analysis in Appendix J.

**Remark 5: Indirect-Rewards Surpass Task-Direct Supervision.** We compare our model with Non-LLM domain-specific models targeting geo-localization task such as Geo-CLIP (Vivanco Cepeda et al., 2023). As shown in Table 1, on scene-diverse IMAGEO-GSS dataset, Geo-R1 surpasses GeoCLIP in all aspects of city-scale and country-scale accuracy, as well as in its prediction error. This indicates that Geo-R1 has demonstrated its robustness across diverse weather conditions, scenarios, and environments.

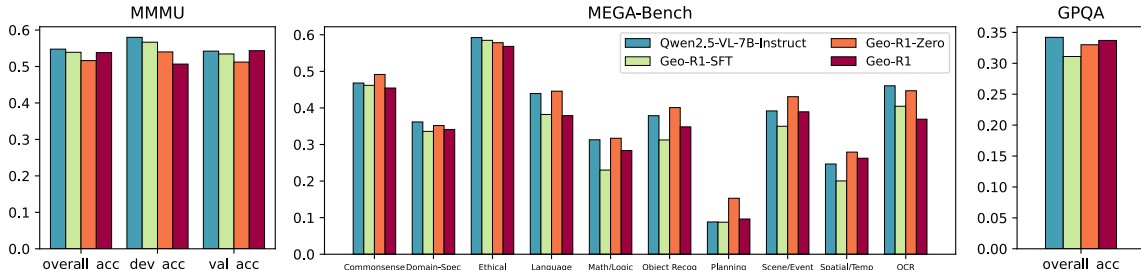

*Figure 7.* Results on non-geospatial general-purpose task benchmarks.

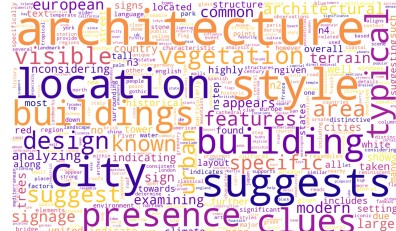

*Figure 8.* Wordcloud when inferring MP16-Reason.

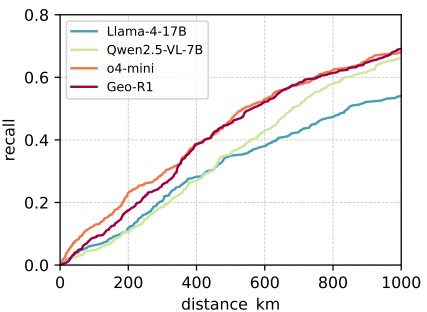

*Figure 9.* Results on RSTeller geolocation.

Furthermore, we compared our model against a series of task-specific models trained specifically for IM2GPS on geolocation's commonly used IM2GPS3K benchmark. As shown in Table 11, we tested Geo-R1 in two modes: (1) without any thinking, directly outputting answers (w/o thinking), and (2) enabling thinking mode before providing responses (w/ thinking). In both modes, the model operates entirely in zero-shot mode. When enabled in thinking mode, the model achieves state-of-the-art performance. Geo-R1's direct output performance is also competitive without thinking. This fully demonstrates that Geo-R1's geolocation capabilities are based on the model's fundamental understanding of geospatial information, rather than overfitting.

### 5.2.3. MP16-REASON: GEOLOCATION VIA REASONING

**Benchmark.** MP16-Reason (Li et al., 2025a) is a dataset build upon ground-view geolocation dataset MP16 (Larson et al., 2017). It features explicit reasoning for geolocation.

**Remark 6: Indirect Rewards Match the Efficacy of Task-Direct Strong Rewards.** Table 3 compares our method against GLOBE-7B (Li et al., 2025a), a state-of-the-art model trained specifically on the MP16-Reason training

set using RL with task-direct strong rewards (i.e., direct supervision on reasoning correctness). Remarkably, Geo-R1 achieves performance nearly identical to GLOBE-7B at all distance precision levels. This result demonstrates that our indirect reward signal, which is derived solely from scalable, proxy-task alignment without seeing any MP16-Reason training data, is sufficient to induce the same level of reasoning capability as strong, task-direct rewards.

**Remark 7: Indirect Rewards Induce Human-Aligned Geospatial Concepts.** Fig. 8 visualizes the wordcloud of Geo-R1's reasoning traces. Despite binary indirect supervision, high-level semantic concepts spontaneously emerge, including "architecture," "vegetation," and "climate." Deductive terms like "analyzing" further confirm active reasoning over passive pattern matching. This qualitative evidence suggests that indirect rewards successfully compress complex physical laws into human-aligned reasoning patterns.

### 5.2.4. RSTELLER: SATELLITE-VIEW GEOLOCATION

**Remark 8: Indirect Rewards Boost Aerial-View Geolocation Capability.** We further consider an out of distribution geolocation task: estimating the location of high-resolution aerial images. We adopt a subset of RSTeller (Ge et al., 2025), which is a data distribution (U.S. Agricultural Land) Geo-R1 has not encountered before. As shown in Fig. 9, Geo-R1 achieves higher recall than the base model and is on-par with o4-mini, indicating that indirect rewards encounter not only ground-view geolocation but also one from aerial perspective. See Appendix K for more details.

### 5.2.5. GEOBENCH-VLM: AERIAL INTERPRETATION

**Benchmark.** GeoBench-VLM (Danish et al., 2025) evaluates fine-grained remote sensing capabilities, including object counting, scene classification, and disaster assessment, solely on high-resolution aerial imagery.

**Remark 9: Indirect-Rewarded Cross-View Pairing Unlocks Aerial-View Reasoning.** Fig. 6 shows that Geo-R1 achieves state-of-the-art zero-shot performance on aerial-exclusive tasks (e.g., Land Use, Disaster Class), surpassing domain specialists like GeoChat (Kuckreja et al., 2024) and EarthDial (Soni et al., 2025). This confirms a critical hy-

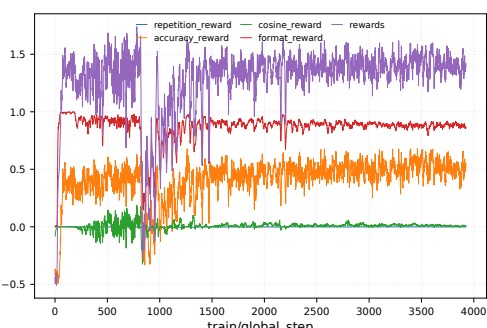 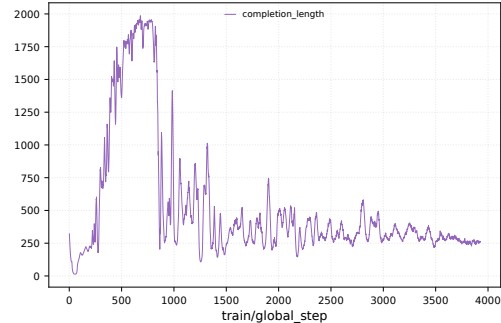

*Figure 10.* GRPO training dynamics. **Left:** rewards dynamic. **Right:** completion length.

*Table 4.* Generalization ability of Geo-R1 on IMAGEO-GSS.

| Model | City Acc. | Ctry. Acc. | Mean Dist. | Med. Dist. |
|---|---|---|---|---|
| InternVL3-8B | 0.146 | 0.512 | 1352.6 | 357.1 |
| + Geo-R1 | **0.313** | **0.723** | **603.2** | **84.3** |
| Qwen2.5-VL-7B | 0.277 | 0.717 | 625.2 | 87.4 |
| + Geo-R1 | **0.327** | **0.815** | **568.3** | **69.4** |

pothesis about indirect rewards: to solve the cross-view pairing proxy task, the model was forced to internalize a robust inverse projection logic that learning to map semantic concepts (e.g., "flooded street") observed on the ground to their abstract aerial counterparts. Therefore, this cross-view training implicitly constructed a powerful aerial image interpreter. Even without explicit supervision on aerial object detection, the indirect reward signal compelled the model to decode aerial features (texture, density, scale) simply to establish reliable correspondences with the ground view.

### 5.3. Preservation of Primitive Abilities

**Remark 10: Geo-R1 Avoids Catastrophic Forgetting.** We find that our post training does not noticeably decrease the performance on the primitive tasks. We evaluate `Geo-R1`, the base model `Qwen2.5-VL-7B`, as well as `Geo-R1-Zero` and `Geo-SFT` on general purpose VLM benchmarks like MEGA-Bench (Chen et al., 2025), GPQA (Rein et al., 2024), and MMMU (Yue et al., 2024). As shown in Fig. 7, `Geo-R1` effectively preserves the base model's capabilities in scientific QA, foundational multimodal understanding, etc. See Appendix L for more details.

Notably, we can observe most slight performance degradation on primitive tasks are brought by SFT (green bar), but not by RLVR (orange bar). This indicates the necessity to carefully control SFT steps to be small to avoid catastrophic forgetting. This also highlights our scaffold SFT's advantages to achieve a good tradeoff to use minimal SFT steps for geospatial reasoning paradigm learning.

### 5.4. Training Dynamics

By observing the model's training dynamics, we identified several noteworthy phenomena, which we remark in this subsection. We describe the model's training dynamics in detail in the Appendix H.

**Remark 11: Geospatial "Aha Moment".** During the RL training, as shown in Fig. 10, we observed that the model's reward reached its first peak around 100 steps. We observe that the model's completion length does not exhibit a convergence trend consistent with the reward over the subsequent period. The model's completion length exhibits a pattern of first decreasing and then increasing, consistent with the "Aha moment" observed in Deepseek-R1 (Guo et al., 2025), while the model accuracy reward continues to rise till convergence. We refer to this as the geospatial "Aha-Moment".

**Remark 12: Outputs Stabilize after Double Ascents.** We observe that the model's behavior stabilizes after two ascents. That is saying, at the begining of the RL training, as the model trained, its outputs became increasingly longer but unstable. The model tends to engage in extensive deliberation, but the content of its deliberations is meaningless or redundant. Then, as shown in Fig. 10 in Appendix H the model's reward collapses after exceeding the maximum output length limit (2048). The model is further trained over the subsequent 500 steps until convergence. We find that the model no longer hit the completion length wall. Meanwhile, the model developed a stable and effective intermediate reasoning process during this double ascents of the rewards.

### 5.5. Geo-R1 Framework Generalization

**Remark 13: The Indirect-Reward Paradigm is Model-Agnostic.** To further verify the universality of our proposed indirect reward and framework, we applied the Geo-R1 post-training paradigm to a different VLM architecture, `InternVL3-8B` (Zhu et al., 2025), and evaluated it on the out-of-distribution IMAGEO-GSS dataset. As reported in Table 4, Geo-R1 yields substantial improvements regardless of the backbone: the post-trained `InternVL3-8B` achieves a city accuracy of 31.26%, more than doubling the base model's 14.62%, while drastically reducing the median distance error from 357.09 km to 84.31 km. While for the in-distribution cross-view pairing task, the Geo-R1 framework improves the `InternVL3-8B`'s accuracy from 20.3% to 84.7%. This demonstrates that Geo-R1 can consistently unlock latent geospatial reasoning capabilities and achieve

performance gain across diverse VLM architectures.

# 6. Conclusion

We validate that indirect rewards derived from cross-view pairing are sufficient to induce robust geospatial reasoning. Geo-R1 empirically proves that optimizing for this verifiable proxy compels models to internalize view-invariant physical laws, bypassing the need for dense supervision. This confirms that "indirect" signals can trigger "strong" zero-shot capabilities, establishing a scalable pathway to unlock latent intelligence within unlabelled archives through verifiable consistency rather than expert labels.

# Impact Statement

This work demonstrates the potential to address diverse societal needs by unlocking robust geospatial reasoning capabilities. By enabling zero-shot interpretation of unfamiliar environments, Geo-R1 has the potential to assist in critical tasks such as disaster response and search and rescue operations, where verifying locations from sparse visual data is essential. Furthermore, the framework's ability to synthesize visual cues offers potential utility for automated urban planning, environmental monitoring, and sociocultural studies. Ultimately, by leveraging massive unlabelled image archives, this approach holds the potential to democratize geospatial analysis, facilitating broader scientific discovery in climate and geographic research.

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

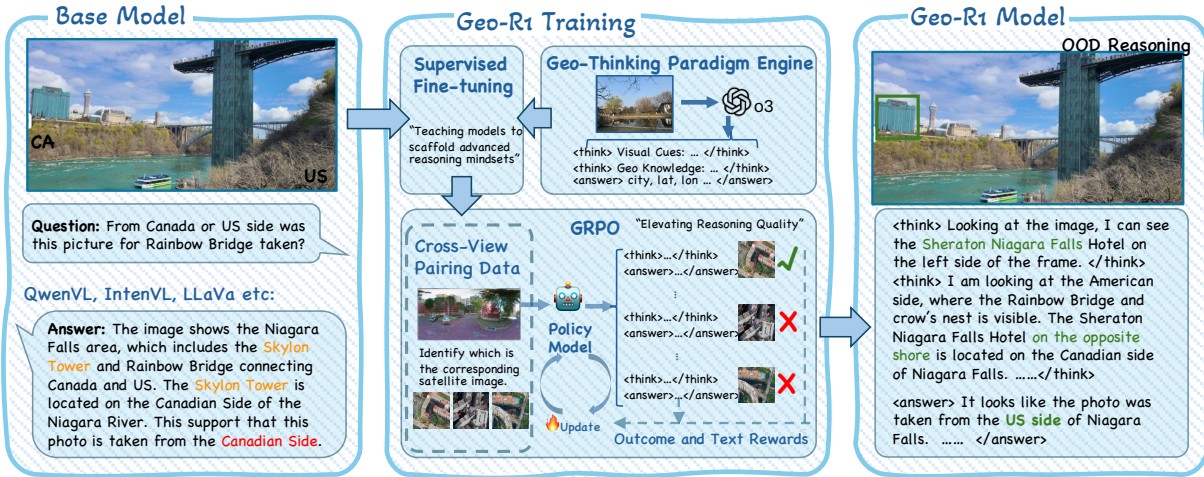

*Figure 11.* Geo-R1 overview. Geo-R1 provide a framework for building geospatial reasoning.

## A. Geo-R1: Scaffolding & Elevating Geospatial Reasoning

As shown in Fig. 11, Geo-R1 is conceptualized as a two-stage methodology engineered to unlock sophisticated geospatial reasoning capabilities in pre-trained VLMs. The approach harmonizes two philosophically distinct stages: (1) an **scaffolding** stage, which leverages small-scale SFT to instill a structured "geo-thinking" paradigm, and (2) an **elevating** stage, which employs larger-scale RLVR to refine the model's reasoning for factual correctness and conciseness through a verifiable proxy task. This two-stage design directly addresses several critical challenges in the field: the absence of innate geospatial reasoning capabilities in general-domain VLMs, the scarcity of expert-annotated reasoning datasets, and the difficulty in formulating a direct, verifiable reward signal for complex, open-ended geospatial reasoning tasks. We discuss the details in Sec. B and Sec. C.

## B. Geo-R1 Stage#1: Geospatial Thinking Scaffolding with SFT

The first stage of the Geo-R1 framework is dedicated to geospatial thinking **scaffolding**, based on the principle that a model must first learn structured, domain-appropriate reasoning skillsets before it can be effectively optimized for accuracy at scale. This SFT phase is carefully designed to scaffold coherent geospatial reasoning, providing the model with an initial foundation of reasoning.

### B.1. Geospatial Thinking Scaffolding

Towards the goal of building a cognitive scaffold for geospatial reasoning, we decide a principle that teaching *domain-generic* reasoning paradigms is more valuable than supervising *question-specific* reasoning and answers, the latter of which can be too diverse for both model learning and data collection at scale. Accordingly, we do not choose to synthesize diverse CoTs for all tasks used in later benchmarks. Instead, we construct a comprehensive geospatial reasoning paradigm and use it to guide our CoT synthesis on a single multi-view reasoning task as shown in Fig. 4:

1. **Visual Cue Identification:** Systematically extract any-view geospatial and semantic features: architectural styles, vegetation/biome, road topology and markings, coastline/river patterns, topography, signage/scripts, and solar cues (sun azimuth/elevation, shadows);

2. **Knowledge Association:** Map cues to geospatial priors (climate bands, cultural/linguistic regions, urban morphology). For example, link tile roofs and narrow alleys to mediterranean Europe, or infer northern winter hemisphere from low solar elevation and shadows;

3. **Evidence Corroboration:** Cross-refer multiple, potentially weak cues across views; check consistency, resolve contradictions, and prefer hypotheses with convergent evidence;

4. **Conclusion Formulation:** Synthesize the corroborated evidence into a concise answer, optionally noting uncertainty when evidence is limited.

Our scaffolding practice differs from traditional SFT philosophy of "cold-starting" VLMs on diversified target tasks just in order to accelerate the reward acquiring process during RL. Instead, we teach the target VLM a unified geospatial reasoning paradigm, which is created using a single template and from a single data source.

Such a design shift brings in multi-fold advantages: (1) It provides similar benefits of regular SFT, including stabilizing early training of RL, enabling RL to rampup faster, and also achieving the goal of teaching generic reasoning behaviors that transfer to diverse downstream tasks; (2) Moreover, the scaffolding SFT design can greatly reduce the SFT task diversity, further leading to reduction in amount of SFT steps needed. This is a key for Geo-R1 to prevent catastrophic forgetting compared to other heavy SFT stages in common post-training frameworks. (3) Lastly, such a unified CoT data acquiring process is very efficient in geospatial domain, which involve much less QA design and collection overheads facing limited geospatial data annotation sources.

### B.2. Geospatial CoT Data Engine

**CoT Synthesis.** We collect images with from cross-view geospatial dataset, CV-Cities (Huang et al., 2025), which contains 223,736 panorama-satellite image pairs with geolocation data. The samples span 16 cities across 13 countries and cover four major daily scenes (city, natural, water, occlusion). As shown in Fig.4, we prompt OpenAI-o3[1] to produce city labels and latitude/longitude for panorama - satellite image pairs, explicitly instructing cross-view analysis following our thinking paradigms: identify salient visual cues across perspectives, perform multi-view feature matching, and integrate pertinent geospatial knowledge. We collect the intermediate reasoning trajectories to form a corpus of 12,646 samples (around 43.6 MB of text). To reduce overfitting during SFT, we set to medium reasoning strengths to generate moderately detailed intermediate CoTs.

**Fact-Check Engine.** We implement the fact-check engine to serve as an automatic verifier that grounds the model's reasoning and final answers against concrete geospatial metadata. It reduces hallucinations, enforces adherence to real-world constraints, and ensures that the reasoning process remains anchored to verifiable spatial facts. The fact-check engine works through inference-based self-refinement (Madaan et al., 2023; Liu et al., 2024a): after o3 generates an reasoning trajectory and tentative answer, the engine cross-validates key outputs, (e.g. predicted city, latitude, and longitude) against curated factual references. If inconsistencies are detected (e.g., mismatched coordinates or unsupported city claims), the system prompts in a new conversation with both CoTs and GTs to refine the reasoning trace to be factually coherent and geographically consistent.

### B.3. Supervised Finetuning

During the SFT training process, we fine-tune the base model (`Qwen2.5-VL-7B`, Bai et al. (2025)) with the synthetic dataset as discussed in Section B.2. The training objective is formulated as a standard autoregressive, next-token prediction. The model is trained to minimize the cross-entropy loss over the target sequence, which is a concatenation of the CoT string ($<$think$>$... $<\backslash$think$>$) and the answer ($<$answer$>$... $<\backslash$answer$>$). We perform full-parameter fine-tuning on the model, including the language backbone, visual tower and multi-modal projector.

## C. Geo-R1 Stage#2: Geospatial Thinking Elevating with RLVR

While the scaffolding stage equips the model with a structured geospatial reasoning paradigm, it does not guarantee factual precision, robustness, or multi-view consistency. To bridge this gap, the **elevating** stage leverages RLVR to improve reasoning quality under verifiable rewards.

### C.1. Weak Supervision for Strong Elevation

The most critical challenge in RLVR lies in designing a good reward task, that can (1) be verified at scale, and (2) motivate high-quality reasoning behaviors. Achieving scalability is hard in geospatial reasoning as the only scalable supervision along with raw images is their metadata. Therefore, the key becomes how to leverage weakly-supervised metadata to create

---

[1]https://openai.com/index/introducing-o3-and-o4-mini/

challenging reasoning tasks.

We propose the proxy RL task: matching a ground-level panoramic image to its corresponding satellite view image with confusers, as shown in Fig. 3. We let the model perform a $k$-way single-choice task. For each panoramic image $I_p$, a set of $k$ satellite image candidates $\{I_s^1, \cdots, I_s^k\}$ is provided. This set contains exactly one correct match and $k$-1 challenging confusers, such as satellite images of nearby but incorrect locations within the same city. The model's objective is to generate high-quality reasoning to help correctly identify the choice of the matching satellite image.

Cross-view pairing is a task that is both challenging to solve and easy to verify. On the one hand, a general-purpose VLM that has not been specifically trained on cross-view pairing performs close to random guess, since satellite images selected from the same city often exhibit highly similar architectural and vegetation styles, as shown in Fig. 3. Moreover, the level of reasoning quality required for this task is extremely difficult to reach through SFT. This is proved in Table 1, for single-choice task with five options, `Qwen2.5-VL-7B` model achieves only 19% accuracy. After undergoing a phase of SFT training and being injected with substantial latent knowledge and positive CoT examples linking cross-view images, the model only gains approximately +4% in performance to 23%, barely outperforming random guesses. Therefore, such reward is nearly impossible to hack unless the model truly learns to identify useful corresponding visual cues efficiently and accurately.

On the other hand, the cross-view pairing task is easy to verify and suits large-scale RLVR training since raw images and metadata are broadly available. Such a challenging but non-hackable reward motivates the model to continuously refine its explorations and elevate the model's reasoning quality, ultimately learning strong geospatial reasoning foundation. In our experiments, we found model under-through RLVR demonstrates much stronger reasoning behaviors to capture and synthesize various types of visual information, including car plates, billboard, texts, cultural elements, tree types, traffic signs, building colors and styles, and even car brands, all serving as the foundation for any out-of-domain generalized geospatial reasoning.

### C.2. Reward Design

To make the model's output align with our preferences for better output, we combine direct, verifiable outcome rewards with light textual shaping:

$$r = \lambda_{\mathrm{acc}}\, r_{\mathrm{acc}} + \lambda_{\mathrm{fmt}}\, r_{\mathrm{fmt}} + \lambda_{\mathrm{len}}\, r_{\mathrm{len}} + \lambda_{\mathrm{rep}}\, r_{\mathrm{rep}}.$$

**Accuracy.** $r_{\mathrm{acc}} = +1$ if the correct $\hat{I}_s = I_s^\star$ is identified, $-0.8$ if $\hat{I}_s \neq I_s^\star$, and $-1$ for non-answers / malformed choices; this yields a dense, calibrated signal while discouraging degenerate refusals.

**Format.** $r_{\mathrm{fmt}} \in \{0, 1\}$ grants credit for emitting the agreed `<think>`/`<answer>` structure.

**Length.** $r_{\mathrm{len}}$ rewards succinct but sufficient justification (encourage thinking, avoid over-thinking). We adopt cosine reward (Yeo et al., 2025) with parameters $r_0^c = 0, r_0^w = -1, r_L^c = 0.5, r_L^w = 0$, punishing overly brief responses when the answer is wrong. See details in Appendix F.

**Repetition.** $r_{\mathrm{rep}} \leq 0$ penalizes token-level pattern loops to mitigate post-training repetition. This mixture encourages disciplined reasoning that is both verifiable and readable.

### C.3. Reinforcement Learning Post-Training

We optimize with Group Relative Policy Optimization (GRPO, Shao et al., 2024). For each prompt we sample $M$ rollouts $\{y^{(m)}\}_{m=1}^M$, compute rewards $\{r^{(m)}\}$, and form group-wise advantages:

$$A^{(m)} = r^{(m)} - \frac{1}{M} \sum_{j=1}^M r^{(j)}. \tag{4}$$

The policy $\pi_\theta$ is updated with a clipped objective and a KL regularizer:

$$\max_{\theta} \ \mathbb{E}\Big[\min\Big(\rho_{\theta}^{(m)} A^{(m)}, \ \mathrm{clip}\big(\rho_{\theta}^{(m)}, 1{-}\epsilon, 1{+}\epsilon\big) A^{(m)}\Big)\Big]$$
$$- \ \beta \, \mathrm{KL}(\pi_{\theta} \, \| \, \pi_{\mathrm{ref}}) \,, \tag{5}$$

where $\rho_{\theta}^{(m)} = \frac{\pi_{\theta}(y^{(m)}|x)}{\pi_{\mathrm{ref}}(y^{(m)}|x)}$ is the likelihood ratio and $\pi_{\mathrm{ref}}$ is a frozen reference model.

**Outcome-Based Reward.** We allow free-form '<think>' before parsing a single final choice in '<answer>'. The outcome reward ensures stable gradients even at small scales, while textual shaping regularizes behavior and prevents verbosity drift. In practice we (i) filter unparseable rollouts via the format check, (ii) anneal, lowering the sampling temperature during training to gradually reduce exploration, and (iii) monitor the KL term to avoid collapsing to the reference or diverging into reward-hacking.

## D. Proof of Theorem 3.1

In this section, we provide a derivation of Theorem 3.1. We prove that under the strict hard-negative constraint, maximizing the RL reward is **strictly equivalent** to maximizing the mutual information with view-invariant features, without relying on approximations.

Let the random variables be defined as follows:

- $Y \in \{1, \ldots, K\}$: The index of the ground truth image within the candidate set $\mathcal{S}$.

- $C$: The reasoning chain generated by the policy $\pi_{\theta}(C|I_g)$.

- $\Phi$: The view-invariant geometric features (e.g., road topology).

- $N$: The nuisance/modality-specific features (e.g., weather, style).

We model the selection process as a Markov Chain where the true identity $Y$ determines the observed features of the target image $I_Y$:

$$Y \to (\Phi_Y, N_Y) \to \text{Observation} \tag{6}$$

We define the "Hard Negative" setting formally. The candidate set $\mathcal{S}$ is constructed such that all candidates (ground truth and confusers) share statistically identical nuisance features.

**Definition 1 (Nuisance Indistinguishability).** The probability distribution of the target index $Y$ is independent of the nuisance features $N_{\mathcal{S}}$ present in the candidate set:

$$P(Y = y \mid N_{\mathcal{S}}) = P(Y = y) = \frac{1}{K}. \tag{7}$$

This implies that nuisance features contain **zero information** regarding the true identity of the target:

$$\mathcal{I}(Y; N_{\mathcal{S}}) = 0 \tag{8}$$

Conversely, the target is uniquely identified by invariant features $\Phi$:

$$H(Y \mid \Phi_{\mathcal{S}}) = 0 \tag{9}$$

### D.1. Derivation Steps

The RL objective is to maximize the expected log-likelihood of the correct target $Y$ given the reasoning chain $C$ and the candidate set $\mathcal{S}$:

$$\mathcal{J}(\theta) = \mathbb{E}\left[\log P(Y \mid C, \mathcal{S})\right] \tag{10}$$

By the definition of Mutual Information $\mathcal{I}(X;Y) = H(Y) - H(Y|X)$, and noting that the prior entropy $H(Y|\mathcal{S}) = \log K$ is a constant with respect to $\theta$, maximizing the likelihood is strictly equivalent to maximizing the mutual information:

$$\arg\max_{\theta} \mathcal{J}(\theta) = \arg\max_{\theta} \Big( \underbrace{H(Y|\mathcal{S})}_{\text{const}} - H(Y \mid C, \mathcal{S}) \Big) = \arg\max_{\theta} \mathcal{I}(C;Y \mid \mathcal{S}) \tag{11}$$

We expand the mutual information term $\mathcal{I}(C;Y \mid \mathcal{S})$. Since $Y$ is fully determined by the tuple of features $(\Phi_Y, N_Y)$ in the ideal generative process, we can analyze the information flow through these latent variables. Using the *Chain Rule for Mutual Information*:

$$\mathcal{I}(C;Y \mid \mathcal{S}) = \mathcal{I}(C; \Phi_Y \mid \mathcal{S}) + \mathcal{I}(C;Y \mid \Phi_Y, \mathcal{S}) \tag{12}$$

Here, the second term $\mathcal{I}(C;Y \mid \Phi_Y, \mathcal{S})$ represents the information $C$ contributes to identifying $Y$ *after* the invariant features $\Phi_Y$ are already known.

We define the residual uncertainty. Given $\Phi_Y$, the only remaining variation in $Y$ that could potentially be explained by $C$ must come from $N_Y$. However, by **Definition 1** (Nuisance Indistinguishability), $N$ is statistically independent of $Y$.

Applying the conditional independence property:

$$P(Y \mid C, \Phi_Y, \mathcal{S}) = P(Y \mid \Phi_Y, \mathcal{S}) \tag{13}$$

This equality holds because $C$ cannot extract information about $Y$ from $N$ that does not exist in $N$ (Data Processing Inequality). Since $N$ is uninformative for $Y$, knowing $C$'s representation of $N$ adds no predictive power over $Y$ once $\Phi$ is known.

Therefore, the conditional mutual information term vanishes strictly:

$$\mathcal{I}(C;Y \mid \Phi_Y, \mathcal{S}) = 0 \tag{14}$$

Substituting Eq. 14 back into Eq. 12, we obtain the strict equality:

$$\mathcal{I}(C;Y \mid \mathcal{S}) = \mathcal{I}(C; \Phi_Y \mid \mathcal{S}) \tag{15}$$

Combining Eq. 11 and Eq. 15, we conclude the derivation:

$$\max_{\theta} \mathcal{J}(\theta) \iff \max_{\theta} \mathcal{I}(C; \Phi_Y \mid \mathcal{S}) \tag{16}$$

This proves that under the hard-negative constraint, the optimization landscape offers **no gradient** for encoding nuisance factors $N$. The reasoning chain $C$ improves the objective function if and only if it increases the mutual information with the view-invariant geometric features $\Phi_Y$. ∎

## E. Theoretical Analysis of Learnability and Generalization

In this section, we extend the information-theoretic framework established in Appendix D to formally prove two critical properties of the proposed Cross-View Alignment task: (1) its optimal difficulty regime for RL training (non-trivial yet solvable), and (2) the mathematical guarantee of out-of-distribution (OOD) generalization using the framework of Invariant Risk Minimization.

### E.1. Proof of Optimal Task Difficulty (The Entropy Gap)

We quantify the difficulty of the task using the *Conditional Entropy* of the target variable $Y$ given the reasoning chain $C$. We show that the Hard-Negative design creates a strict "Entropy Gap" between nuisance-based policies and geometry-based policies.

**Lower Bound (The Non-Triviality Condition).**   Consider a "Shortcut Policy" $\pi_{shortcut}$ that generates a reasoning chain $C_N$ relying exclusively on nuisance factors (e.g., texture, weather, vegetation style). From Definition 1 (Appendix A), nuisance factors $N$ are statistically independent of the target $Y$.

The uncertainty of the target given such a policy is defined as:

$$\mathcal{H}(Y \mid C_N) = -\sum_{y=1}^{K} P(y \mid C_N) \log P(y \mid C_N) \tag{17}$$

Since $C_N$ is a function of $N(\mathcal{S})$ and $Y \perp N(\mathcal{S})$, knowing $C_N$ provides no information about $Y$. Thus, the posterior probability remains equal to the prior. Given that the target $Y$ is uniformly distributed over the $K$ candidates:

$$P(y \mid C_N) = P(y) = \frac{1}{K} \tag{18}$$

Substituting this into the entropy definition:

$$\begin{aligned}
\mathcal{H}(Y \mid C_N) &= -\sum_{y=1}^{K} \frac{1}{K} \log \left( \frac{1}{K} \right) \\
&= -K \cdot \left[ \frac{1}{K} (-\log K) \right] \\
&= \log K
\end{aligned} \tag{19}$$

Therefore, any policy relying on superficial features faces maximum uncertainty ($\log K$). This proves the task is **non-trivial** and cannot be solved by "guessing" or "pattern matching" nuisance correlations.

**Upper Bound (The Solvability Condition).** Consider an "Optimal Policy" $\pi^*$ that generates a reasoning chain $C_\Phi$ explicitly capturing the view-invariant geometric semantics $\Phi$. We assume the physical world satisfies Geometric Consistency: there exists a deterministic bijective mapping $M$ (e.g., projective transformation) such that the ground-level geometry $\Phi(I_g)$ corresponds uniquely to the satellite geometry $\Phi(I_s^*)$.

$$Y = \arg\min_{k} \text{Distance}(M(\Phi(I_g)), \Phi(I_s^k)) \tag{20}$$

Under this assumption, knowledge of $C_\Phi$ fully determines $Y$:

$$P(Y = y \mid C_\Phi) = \begin{cases} 1 & \text{if } y = y^* \\ 0 & \text{otherwise} \end{cases} \tag{21}$$

Calculating the entropy for the optimal policy:

$$\begin{aligned}
\mathcal{H}(Y \mid C_\Phi) &= -\sum_{y=1}^{K} P(y \mid C_\Phi) \log P(y \mid C_\Phi) \\
&= -(1 \cdot \log 1 + 0 + \dots) \\
&= 0
\end{aligned} \tag{22}$$

Therefore, the task is **solvable**.

**The Reasoning Margin.** We define the *Reasoning Margin* $\Delta$ as the entropy reduction required to solve the task:

$$\Delta = \mathcal{H}(Y \mid C_N) - \mathcal{H}(Y \mid C_\Phi) = \log K \tag{23}$$

This specific margin $\log K$ represents the information gain strictly attributable to reasoning about $\Phi$. The RL optimization process is effectively maximizing this margin.

### E.2. Proof of OOD Generalization via Causal Invariance

We now prove that minimizing the loss on the training distribution guarantees performance on unseen test distributions (OOD), provided the model learns $\Phi$. We use the language of **Invariant Risk Minimization (IRM)**.

Let $\mathcal{E}$ be the set of all environments (domains), where $e \in \mathcal{E}$ represents a specific city or region (e.g., $e_{train} = $ New York, $e_{test} = $ Tokyo). Data generation in any environment $e$ follows a Structural Causal Model (SCM):

- $N \leftarrow f_N(e)$ (Nuisance features depend on the environment, e.g., architecture style).

- $\Phi \leftarrow f_\Phi(\text{Physics})$ (Geometric features depend on immutable physical laws).

- $Y \leftarrow f_Y(\Phi)$ (The label depends *only* on geometric correspondence).

The core axiom of geospatial reasoning is that the laws of perspective geometry are universal. Thus, the conditional probability of the label $Y$ given the invariant features $\Phi$ is **stable** across all environments:

$$\forall e_1, e_2 \in \mathcal{E}: \quad P_{e_1}(Y \mid \Phi) = P_{e_2}(Y \mid \Phi) = P(Y \mid \Phi) \tag{24}$$

However, the relationship between nuisance features and the label is unstable (or uniformly random in our hard-negative setup):

$$P_e(Y \mid N) \text{ varies or is undefined across } e \tag{25}$$

Let the optimal classifier based on reasoning chain $C$ be $h(C)$. The risk (expected error) in environment $e$ is:

$$R_e(h) = \mathbb{E}_{(x,y) \sim e}[\mathcal{L}(h(C(x)), y)] \tag{26}$$

If the reasoning chain $C$ captures $\Phi$ (as proven necessary in Appendix D), then $C \to \Phi$. Substituting the invariance assumption (Eq. 26):

$$
\begin{aligned}
R_e(h) &= \sum_\Phi P_e(\Phi) \sum_y P(y \mid \Phi) \mathcal{L}(h(\Phi), y) \\
&= \mathbb{E}_{\Phi \sim P_e}[\text{Risk}(h \mid \Phi)]
\end{aligned}
\tag{27}
$$

Here, $\text{Risk}(h \mid \Phi)$ is the pointwise error for a specific geometric configuration. Importantly, this pointwise risk depends **only** on the universal conditional $P(Y|\Phi)$, not on the environment $e$.

**Zero-Shot Generalization Guarantee.**   Consider the generalization gap between training environment $e_{train}$ and test environment $e_{test}$. Since $C$ encodes only $\Phi$ (due to the hard-negative bottleneck), the classifier $h$ mimics the true causal mechanism $P(Y|\Phi)$.

$$h(C) \to P(Y \mid \Phi) \tag{28}$$

The risk difference depends solely on the shift in the marginal distribution of geometry $P(\Phi)$ (Covariate Shift), but **not** on Concept Shift (change in rules):

$$R_{test} - R_{train} = \int \left( P_{test}(\Phi) - P_{train}(\Phi) \right) \cdot \text{Risk}(h \mid \Phi) \, d\Phi \tag{29}$$

Crucially, because the underlying function $f_Y(\Phi)$ (geometric matching) is learned perfectly in $e_{train}$, the term $\text{Risk}(h \mid \Phi)$ approaches 0 for all valid $\Phi$.

$$\text{If } h \to P(Y|\Phi), \text{ then } \text{Risk}(h \mid \Phi) \to 0 \implies R_{test} \to 0 \tag{30}$$

Therefore, by forcing the model to learn the invariant causal mechanism $\Phi$ during training, the model achieves low risk on any OOD environment $e_{test}$ where the physical of geometry hold, regardless of shifts in nuisance distributions $P(N)$.  ■

## F. Cosine Rewards

We use a cosine-shaped length reward (Yeo et al., 2025) to encourage *succinct but sufficient* reasoning: reward increases smoothly with the number of generated reasoning tokens until a target cap and then plateaus (no incentive to over-think).

**Setup.**   Let $n \in \mathbb{N}$ be the number of reasoning tokens produced before the final answer token. Let $n_{\min} \geq 0$ be the minimum length after which we start rewarding, and $n_L > n_{\min}$ be the target cap beyond which extra tokens bring no additional length reward. Define the normalized clipped length

$$s(n) = \text{clip}\left( \frac{n - n_{\min}}{n_L - n_{\min}}, 0, 1 \right), \quad \phi(s) = \tfrac{1}{2}(1 - \cos(\pi s)) \in [0, 1].$$

Here $\phi(\cdot)$ is monotonically increasing, has zero slope at both ends, and provides a smooth rise without sharp incentives to chase the cap, we set the $n_{min} = 0$ and $n_L = 2048$ in the Geo-R1 training.

Let $y \in \{c, w\}$ denote whether the final answer is correct ($c$) or wrong ($w$). Given boundary rewards $\{r_0^y, r_L^y\}$ (at $s = 0$ and $s = 1$ respectively), the cosine length reward is

$$r_{\text{len}}(n, y) \;=\; r_0^y \;+\; \left(r_L^y - r_0^y\right) \phi\!\left(s(n)\right). \tag{A.1}$$

Intuitively, $r_0^y$ controls how we treat very brief responses, while $r_L^y$ sets the maximum bonus once a sufficient justification is reached.

**Instantiated parameters.**  In our Geo-R1 training case, we use

$$r_0^c = 0, \qquad r_0^w = -1, \qquad r_L^c = 0.5, \qquad r_L^w = 0. \tag{A.2}$$

Thus, (i) a very short response that is *wrong* receives a negative signal ($-1$), penalizing "guessing and quitting"; (ii) once sufficient length is reached, a *correct* response gets a modest bonus ($+0.5$), while a *wrong* response receives no additional bonus, avoiding incentives to "pad" incorrect reasoning; and (iii) beyond $n_L$ there is no further gain, discouraging over-thinking.

Equation (A.1) is an episodic scalar reward added to other task terms (e.g., accuracy, format). Let $\lambda_{\text{len}} \geq 0$ be a weight; the total reward is

$$R \;=\; R_{\text{task}} \;+\; \lambda_{\text{len}}\, r_{\text{len}}(n, y).$$

We tune $\lambda_{\text{len}}$ on held-out tasks; $n_{\text{min}}$ and $n_L$ are hyperparameters tied to the allowed rationale budget (e.g., $n_{\text{min}}$ for ignoring boilerplate, $n_L$ near the per-sample token cap).

**Properties.**  (i) **Monotone & bounded:** $r_{\text{len}}$ increases smoothly from $r_0^y$ to $r_L^y$ as $n$ grows from $n_{\text{min}}$ to $n_L$ and is constant thereafter. (ii) **Short-penalty asymmetry:** with (A.1)–(A.1) we penalize short *wrong* answers while not penalizing short *correct* ones, aligning incentives toward concise correctness. (iii) **No incentive to pad:** because $\phi(1) = 1$ and is flat beyond $n_L$, longer-than-needed rationales do not increase reward.

# G. Training Details

## G.1. Supervised Fine-tuning

We use the LLama-Factory (Zheng et al., 2024) for the supervised fine-tuning. We first conduct supervised fine-tuning on the multimodal backbone using the `Qwen/Qwen2.5-VL-7B-Instruct` (Bai et al., 2025) model as initialization. The model is trained in full fine-tuning mode without freezing any modality-specific components. The maximum input sequence length is set to 131,072 tokens, and up to 10M samples are used for training. Optimization is performed with a cosine learning rate scheduler, peak learning rate of $1.0 \times 10^{-6}$, and warmup ratio of 0.1. Each GPU processes a batch size of 1, and we accumulate gradients for 2 steps. Training is conducted for 2 epochs with bfloat16 precision. Key hyperparameters are summarized in Table 5.

## G.2. GRPO-based Reinforcement Learning

After SFT, we further optimize the model using Group Relative Policy Optimization (Shao et al., 2024). We employ the VLM-R1 (Shen et al., 2025) as the training framework. Training is launched with `torchrun` on 8 A100 GPUs (single node). We employ DeepSpeed ZeRO-3 for memory-efficient distributed optimization. Each GPU uses a per-device batch size of 4, with gradient accumulation of 2 steps, yielding an effective batch size of $4 \times 2 \times 8 = 64$ prompts per update. For each prompt, the model generates 8 candidate completions, resulting in 512 generations per update. The maximum completion length is set to 2048 tokens. Reward functions include `accuracy`, `format`, `length`, and `repetition`, with a KL/entropy regularization coefficient $\beta = 0.04$. We adopt FlashAttention-2, gradient checkpointing, and mixed precision (bfloat16) to improve efficiency. GRPO-specific hyperparameters are summarized in Table 6, 7, and 8.

*Table 5.* Key training hyperparameters in SFT stage of Geo-R1.

| Parameter | Value |
|---|---|
| Fine-tuning type | Full |
| Max input length | 131072 |
| Max samples | 10M |
| Batch size (per device) | 1 |
| Gradient accumulation steps | 2 |
| Learning rate | $1.0 \times 10^{-6}$ |
| Epochs | 2.0 |
| Scheduler | Cosine |
| Warmup ratio | 0.1 |
| Precision | bfloat16 |
| DeepSpeed Config | ZeRO-2 |
| Freeze Vision Tower | False |
| Freeze Multi-Modal Projector | False |

*Table 6.* System and parallel configuration for GRPO training.

| Item | Setting |
|---|---|
| GPUs per node | 4/8 |
| Nodes | 1 |
| Total GPUs | 4/8 |
| Precision | bfloat16 |
| Attention kernel | FlashAttention-2 |
| Gradient checkpointing | Enabled |
| DeepSpeed Config | ZeRO-3 |

*Table 7.* Training schedule and bookkeeping.

| Item | Setting |
|---|---|
| Epochs | 2 |
| Per-device batch size | 4 |
| Gradient accumulation | 2 |
| *Effective prompt batch / update* | $4 \times 2 \times 8 = 64$ |
| Logging interval | 1 |
| Max completion length | 2048 tokens |

*Table 8.* GRPO-specific configuration.

| Item | Setting |
|---|---|
| Generations per prompt | 8 |
| *Total generations / update* | $64 \times 8 = 512$ |
| Reward functions | `accuracy, format, length, repetition` |
| KL/entropy coefficient | $\beta = 0.04$ |

During the RL phase, we adopt the following system prompt:

```
"A conversation between User and Assistant. The user asks a question, and the
    Assistant solves it. The assistant first thinks about the reasoning process in
    the mind and then provides the user with the answer. The reasoning process and
    answer are enclosed within <think> </think> and <answer> </answer> tags,
    respectively, i.e., <think> reasoning process here </think><answer> answer here
    </answer>"
```

A data sample is defined as:

```
{"id": 1, "image": ["cv_cities_16k/barcelona/pano_img/--0eE3ZmREVxVXH_oIeIqw.jpg",
    "cv_cities_16k/barcelona/sat_img/--0eE3ZmREVxVXH_oIeIqw.jpg",
    "cv_cities_16k/barcelona/sat_img/1hfQgX1jGYsXaP74MfLSKQ.jpg",
    "cv_cities_16k/barcelona/sat_img/1plY2fbvDkM9yadGq_edzw.jpg",
    "cv_cities_16k/barcelona/sat_img/-cM5TsqoZcV-kYlxxOARBA.jpg",
    "cv_cities_16k/barcelona/sat_img/2iA9_BNIeO3XZgbLamEbPA.jpg"], "conversations":
```

```
[{"from": "human", "value": "<image><image><image><image><image><image> You are shown
one ground-level panorama and five satellite views labeled as A, B, C, D, and E.
Exactly one satellite image depicts the same location. Identify the correct satellite
image. Think step by step, you can generate multi <think> </think> box, bound your
each thinking step with a box. Respond with a single choice A-E in <answer>
</anwser>."}, {"from": "gpt", "value": "A"}]]
```

## H. Traning Dynamics

We show here the policy evolution during GRPO training, aligning with the quantitative trends shown in Figs. 12-21. We report the overall return and dispersion, component-wise rewards (accuracy, repetition, format, and length), optimization diagnostics (loss and gradient norms), and the behavior of completion lengths.

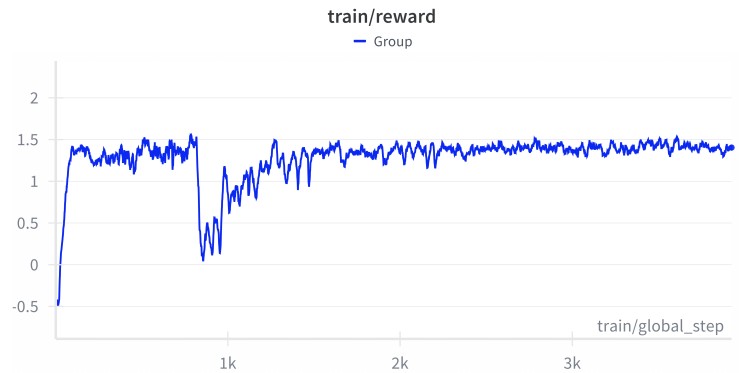

*Figure 12.* Reward dynamic during GRPO training.

### H.1. Overall reward and dispersion

**Overall reward.** As shown in Fig. 12, We observe a rapid rise in average reward at the beginning, followed by a brief stabilization, a secondary climb, and then a steady plateau. The first prominent peak appears within the early updates and matches the "geospatial *aha-moment*" described in the main text: the policy starts to assemble consistently useful spatial cues before settling into a higher-reward regime.

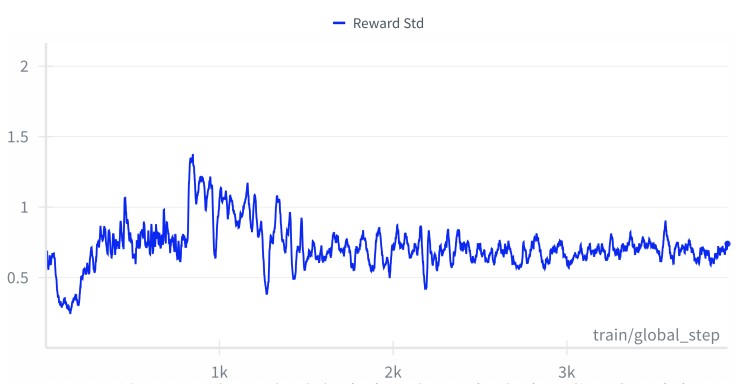

*Figure 13.* Reward standard deviation dynamic during GRPO training.

**Reward dispersion. (Fig. 13)** The within-batch standard deviation is high in the exploratory phase—reflecting diverse and unstable reasoning paths—and gradually contracts as decoding temperature is annealed and format filtering becomes effective. Short, local upticks in variance coincide with exploration boosts or scheduler changes.

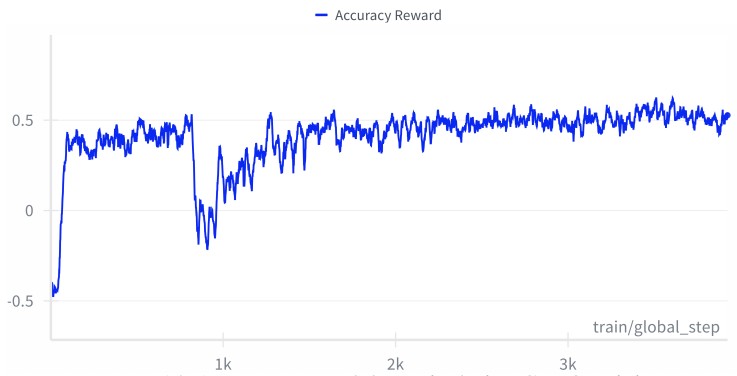

*Figure 14.* Accuracy reward dynamic during GRPO training.

## H.2. Component-wise rewards

**Accuracy reward** ($r_{\mathbf{acc}}$). The mean of $r_{\mathrm{acc}}$ increases monotonically and saturates near the end of training (Fig. 14). We assign a positive credit to correct predictions and a negative credit to incorrect or unparseable outputs, which gives a dense, calibrated learning signal while discouraging "no-answer" degeneracy.

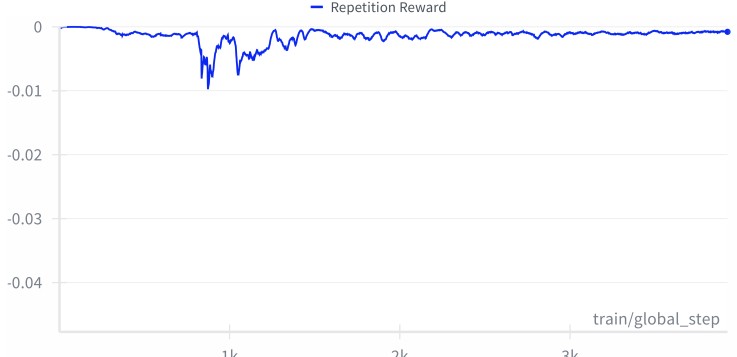

*Figure 15.* Repetition reward dynamic during GRPO training.

**Repetition reward** ($r_{\mathbf{rep}} \leq 0$). The magnitude of the repetition penalty declines toward zero over time, indicating that the policy sheds looped phrases and mechanical echoing, and converges to more concise chains of thought. (Fig. 15)

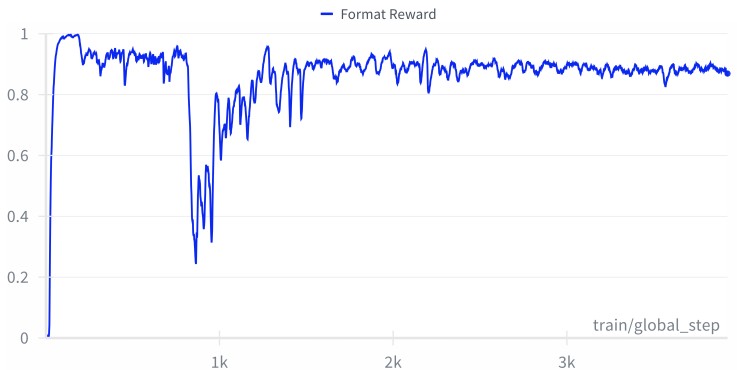

*Figure 16.* Format reward dynamic during GRPO training.

**Format reward** ($r_{\mathbf{fmt}} \in \{0, 1\}$). The fraction of format-compliant generations rises quickly to near-saturation (Fig. 16) once the `<think>...</think><answer>...</answer>` structure is enforced. This stabilizes parsing and downstream evaluation and reduces label noise from ill-formed outputs.

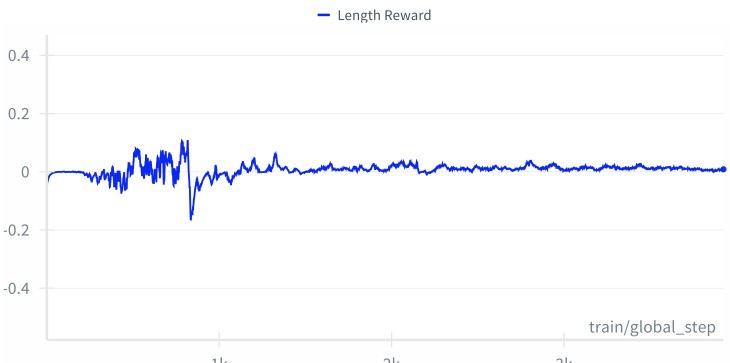

*Figure 17.* Length (cosine) reward dynamic during GRPO training.

**Length / cosine reward** ($r_{\text{len}}$). Empirically (Fig. 17), $r_{\text{len}}$ increases early, then plateaus; when the policy temporarily over-extends to the cap, the net return can dip, prompting a stable reversion to concise-but-sufficient chains.

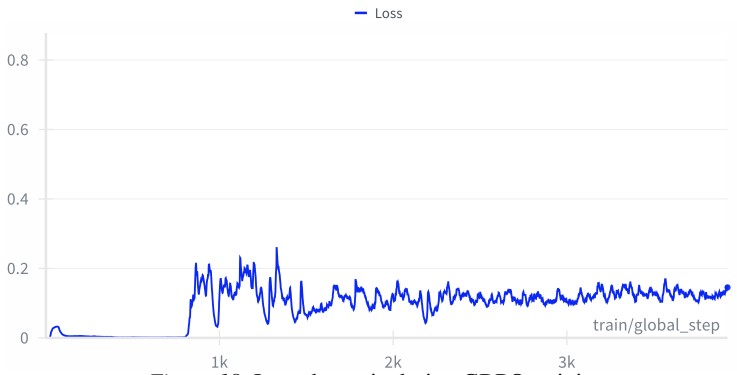

*Figure 18.* Loss dynamic during GRPO training.

### H.3. Optimization diagnostics

**Loss (Fig. 18).** The training loss decreases and then stabilizes, indicating that the policy does not exploit spurious reward loopholes but instead converges around the reference policy under the KL constraint.

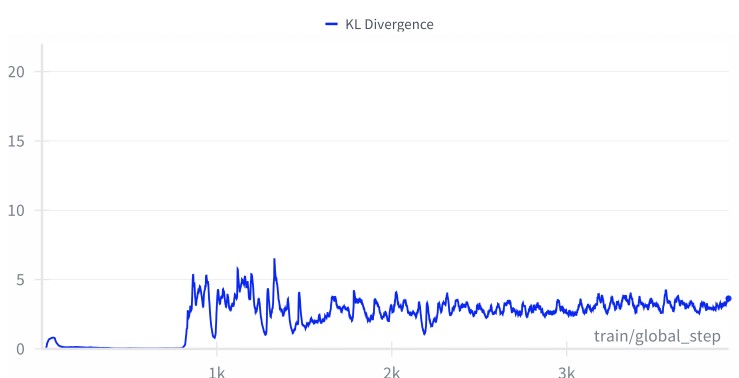

*Figure 19.* KL Divergence dynamic during GRPO training.

**KL Divergence (Fig. 19).** KL divergence fluctuated after encountering the completion length wall and subsequently remained stable, indicating that the model has undergone certain changes relative to the original distribution, but overall remains within a controllable range.

**Gradient norm (Fig. 20).** We observe several early spikes (coinciding with shifts in accuracy/length/format trade-offs),

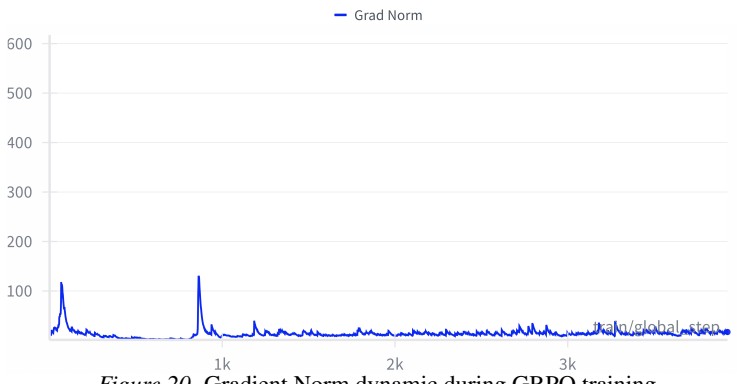

*Figure 20.* Gradient Norm dynamic during GRPO training.

followed by clear stabilization. In practice, large-batch sampling with efficient memory partitioning (e.g., ZeRO) and fast attention kernels keep updates well-behaved.

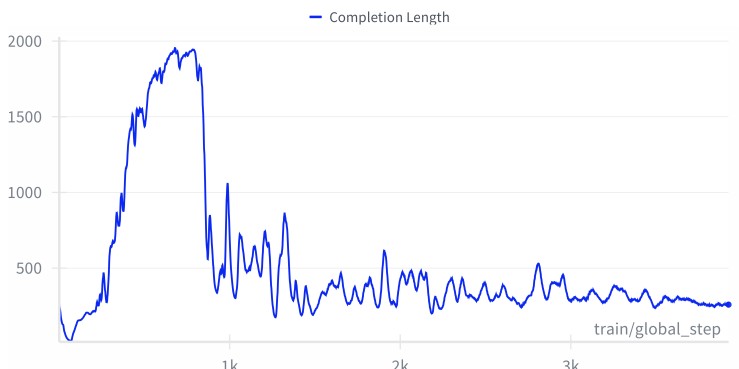

*Figure 21.* Completion length dynamic during GRPO training.

### H.4. Completion length behavior (Fig. 21)

Completion lengths follow a "grow → touch-cap → recede → stabilize" trajectory. In the exploratory phase, the model often hits the 2048-token limit, which, combined with the length/repetition shaping, lowers net returns and nudges the policy toward more compact and more accurate solutions. The stabilized regime features shorter completions that correlate with higher accuracy and lower dispersion.

## I. GeoChain Results

The GeoChain (Yerramilli et al., 2025) dataset is a large-scale benchmark designed to evaluate step-by-step geographic reasoning in multimodal large language models (MLLMs). Built on 1.46 million Mapillary street-level images, it pairs each image with a 21-step chain-of-thought (CoT) sequence, resulting in over 30 million question–answer pairs. These questions progressively guide models from coarse reasoning (e.g., hemisphere, continent) to fine-grained tasks such as city-level identification and predicting precise latitude–longitude coordinates. To support detailed analysis, the dataset includes semantic segmentation maps with 150 visual classes and a locatability score that quantifies how identifiable a location is from visual cues, allowing images to be stratified into Easy, Medium, and Hard difficulty tiers. A curated subset, GeoChain Test-Mini, contains 2,088 diverse and high-quality images for focused evaluation. Overall, GeoChain provides a structured, diagnostic framework that highlights model strengths and weaknesses across visual, spatial, cultural, and geolocation reasoning categories.

We selected 13 subproblems from GeoChain to validate the model's geospatial performance. Because these 13 questions have high-quality annotations. The description of these questions can be seen in Table 9. The subproblems in this dataset are highly challenging. We extract a subset of 800 volumes to validate the model's accuracy on these problems. The results are shown in Table 10 and Fig. 1.

*Table 9.* Selected questions from GeoChain.

| Index | Question | Category | Difficulty |
|---|---|---|---|
| 0 | Would you say this location is near the Equator? | Geographical | Easy |
| 1 | Does this location seem to be close to the Poles? | Geographical | Easy |
| 2 | Is this place located in the Northern Hemisphere? | Geographical | Easy |
| 3 | Which continent best describes where this location is? (7 continents: North America/South America/Europe/Africa/Asia/Oceania/Antarctica) | Geographical | Easy |
| 4 | Is this place near coast? | Terrain/Environmental | Medium |
| 5 | Does this location appear to be an island? | Terrain/Environmental | Medium |
| 6 | Is this place located in a desert region? | Terrain/Environmental | Easy |
| 7 | Does this location seem to be in a mountainous or hilly region? | Terrain/Environmental | Easy |
| 8 | Does this place look like a big city? | Sociocultural | Easy |
| 9 | Would you classify this place as a small town? | Sociocultural | Medium |
| 10 | What language(s) are most likely spoken at this place? | Sociocultural | Hard |
| 11 | Can you name the state or province this place belongs to? | Geolocation | Hard |
| 12 | What is the name of the city, town, or village seen here? | Geolocation | Hard |

*Table 10.* Results on GeoChain subproblems.

| Index | Qwen2.5-VL-7B | Geo-SFT | Geo-R1-Zero | Geo-R1 |
|---|---|---|---|---|
| 0 | 86.75 | 85.75 | 91.75 | 91.50 |
| 1 | 73.00 | 82.50 | 93.50 | 98.875 |
| 2 | 55.75 | 65.75 | 87.75 | 97.75 |
| 3 | 83.75 | 82.75 | 81.75 | 98.125 |
| 4 | 57.25 | 59.25 | 63.25 | 64.75 |
| 5 | 82.50 | 81.50 | 99.00 | 100.00 |
| 6 | 83.25 | 81.25 | 90.25 | 92.00 |
| 7 | 94.25 | 91.25 | 84.25 | 96.75 |
| 8 | 6.625 | 13.625 | 31.625 | 40.25 |
| 9 | 61.50 | 63.50 | 22.50 | 77.75 |
| 10 | 5.50 | 6.50 | 41.50 | 67.375 |
| 11 | 6.25 | 13.25 | 43.25 | 57.75 |
| 12 | 4.50 | 15.50 | 25.625 | 64.75 |

## J. IMAGEO-Bench Results

IMAGEO-Bench is a standardized benchmark for image geolocalization with vision-language models that emphasizes transparency, structure, and real-world diversity. It unifies input–output format via a constrained JSON schema requiring step-by-step visual reasoning (evidence from landmarks, text/signage, cultural cues, and spatial context) together with a predicted address, latitude/longitude, and confidence. The suite spans three complementary datasets—a globally distributed street-level set, a U.S. points-of-interest set, and a private held-out collection—covering outdoor/indoor scenes, urban–suburban variety, and broad geographic coverage to probe generalization and bias. The protocol disallows external tools and embedded GPS during inference to ensure comparability, and it provides reproducible scripts plus multi-granularity metrics (parsability, country/state/city correctness, and great-circle distance) alongside efficiency reporting (token usage/cost). Together, IMAGEO-Bench offers an interpretable, diagnostics-friendly testbed for studying how models extract geospatial cues and where they succeed or fail to generalize across regions and scene types.

We tested the model's comprehensive geolocation reasoning capabilities on two datasets within IMAGEO-Bench: the global dataset-GSS with 6152 samples and the U.S.-wide dataset-UPC with 2928 samples.

As shown in Table 12 and Table 13, Geo-R1 achieves state-of-the-art performance among open-source models on both global-scale and US-scale geolocation tasks. The Geo-R1 model with 7 billion parameters can even outperform models with 90 billion parameters. We observed that Llama-3.2-90B exhibits exceptionally strong coordinate prediction capabilities. This is attributed to its extremely high refusal rate, where it often declines to provide answers for uncertain queries. Consequently, the number of usable responses parsed is minimal, which we do not consider desirable.

The accuracy rates reported in in Table 12 and Table 13 are based on all identifiable responses. Geo-R1 achieved an

*Table 11.* Performance comparison across different geographic localization thresholds.

| Method | Street
1 km | City
25 km | Region
200 km | Country
750 km | Continent
2500 km |
|---|---|---|---|---|---|
| [L]kNN, $\sigma = 4$ (Vo et al., 2017) | 7.2 | 19.4 | 26.9 | 38.9 | 55.9 |
| PlaNet (Weyand et al., 2016) | 8.5 | 24.8 | 34.3 | 48.4 | 64.6 |
| CPlaNet (Seo et al., 2018) | 10.2 | 26.5 | 34.6 | 48.6 | 64.6 |
| ISNs (Muller-Budack et al., 2018) | 10.5 | 28.0 | 36.6 | 49.7 | 66.0 |
| Translocator (Pramanick et al., 2022) | 11.8 | 31.1 | 46.7 | 58.9 | 80.1 |
| GeoDecoder (Clark et al., 2023) | 12.8 | 33.5 | 45.9 | 61.0 | 76.1 |
| GeoCLIP (Vivanco Cepeda et al., 2023) | 14.1 | 34.5 | 50.6 | 69.7 | 83.8 |
| Qwen2.5-VL-7B-Instruct | 5.0 | 26.0 | 46.1 | 64.8 | 77.5 |
| Geo-R1 (w/o. thinking, zero-shot) | 13.2 | 27.2 | 49.0 | 68.2 | 81.4 |
| Geo-R1 (w/. thinking, zero-shot) | **14.2** | **41.3** | **58.4** | **73.1** | **85.1** |

*Table 12.* Test results on IMAGEO-GSS dataset.

| model | city_accuracy | country_accuracy | mean_distance_km | median_distance_km |
|---|---|---|---|---|
| Llama-3.2-11B | 0.140930 | 0.666941 | 797.432896 | 66.563253 |
| Llama-4-17B | 0.256990 | 0.699935 | 840.291329 | 127.015201 |
| Llama-3.2-90B | 0.263459 | 0.668849 | 382.892011 | 15.740222 |
| Qwen2.5-VL-7B | 0.277271 | 0.716517 | 625.207248 | 87.365986 |
| Qwen2.5-VL-32B | 0.306242 | 0.727081 | 780.872273 | 94.564417 |
| Claude-3.5-haiku | 0.306525 | 0.745076 | 568.169894 | 68.827582 |
| o3 | 0.419769 | 0.886685 | 288.075326 | 8.207232 |
| GPT-4o | 0.278131 | 0.786421 | 683.965141 | 88.356123 |
| InternVL3-8B | 0.146229 | 0.511513 | 1352.624585 | 357.089196 |
| Geo-R1 | 0.327264 | 0.814664 | 568.322859 | 69.400873 |

identification success rate exceeding 99%. This implies that the actual performance gap between Geo-R1 and other open-source LLMs is significantly larger than what the IMAGEO Benchmark data reveals, particularly considering that Llama-3.2-90B only responded to instructions in about 46% of cases..

As shown in Fig 22, 23, 24, 25, Geo-R1 generally exhibits higher confidence in its own answers. We observe that the 32B model of Qwen2.5-VL demonstrates stronger benchmark performance than the 7B model, suggesting that training larger benchmark models using the Geo-R1 framework may yield a more robust geospatial reasoning model.

## K. Aerial image geolocation

As an additonal OOD task, we consider aerial image geolocation. While there exists extensive ground view and cross-view (ground view + satellite or aerial) geolocation literature, there are no current benchmarks for geolocating aerial images. For our evaluation we use a small subset of US National Agriculture Imagery Program (NAIP) aerial imagery (490 images total) from (Ge et al., 2025). The aerial images are drawn evenly from across the United States. The image resolution is $448 \times 448$, with a ground sample distance of 0.6 meter per pixel. See Fig. 26 for some examples of the NAIP images used in our evaluation. Many of the images are very challenging, and we did not expect the models to achieve high accuracy at small range.

We employ a CoT prompt to elicit image geolocations from the VLMs. The same prompt is used to evaluate all models:

```
You are shown one aerial image. Provide your best guess of the location on Earth depicted
    by the image. Think step by step, you can generate multi <think> </think> box, bound
    each thinking step with a box. Respond with your answer in (latitude, longitude)
    coordinate tuple, accurate to 4 decimal places in <answer> </answer>. i.e. <answer>
    (lat, lon) </answer>.
```

Given the model response location and ground truth, we calculate the great-circle distance (Haversine) in kilometers, to obtain geolocation error. In our configuration, the models can return image locations from anywhere on Earth. We consider recall for distances less than 1000 km, as considering larger ranges is not practical on a US scale.

We show the recall rate at different distances. As shown in Fig. 9 and Table 14, we show that our Geo-R1 model can on-par the advanced close-source reasoning model o4-mini. Our model achieved significantly better performance than Llama-4-17B and the base model.

*Table 13.* Test results on IMAGEO-UPC dataset.

| model | city_accuracy | state_accuracy | mean_distance_km | median_distance_km |
|---|---|---|---|---|
| Llama-3.2-11B | 0.033194 | 0.189310 | 955.537907 | 353.217494 |
| Llama-4-17B | 0.090444 | 0.248175 | 1217.486267 | 534.276735 |
| Llama-3.2-90B | 0.108540 | 0.239756 | 706.838244 | 162.954925 |
| Qwen2.5-VL-7B | 0.070673 | 0.185478 | 1411.940635 | 862.672667 |
| Qwen2.5-VL-32B | 0.083333 | 0.221610 | 1163.978083 | 775.544569 |
| Claude-3.5-haiku | 0.082572 | 0.300048 | 697.114125 | 258.685726 |
| o3 | 0.239331 | 0.457645 | 662.684007 | 214.273640 |
| Geo-R1 | 0.101602 | 0.284631 | 840.645108 | 468.950354 |

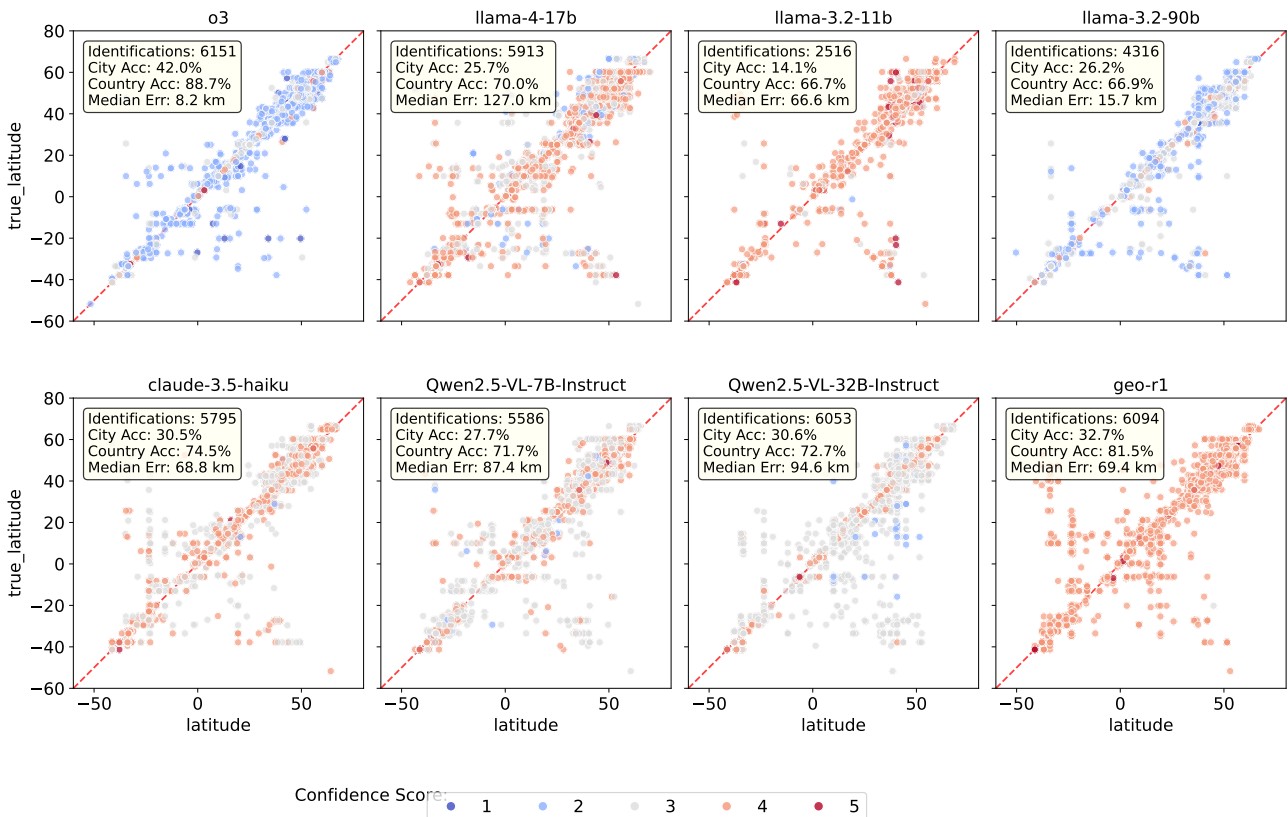

*Figure 22.* Latitude analysis of IMAGEO-GSS

*Table 14.* Model recall as a function of great circle distance threshold, for small subset of RSTeller aerial data (474 images).

| Method | 1 km | 25 km | 200 km | 750 km | 2500 km |
|---|---|---|---|---|---|
| GPT-o4-Mini | 0.0 | 4.6 | 23.0 | 60.2 | 86.4 |
| Geo-R1 | 0.0 | 1.1 | 17.6 | 59.7 | 86.9 |
| Qwen-2.5-VL-7B | 0.0 | 1.9 | 10.5 | 55.2 | 88.9 |

# L. General VLM Tasks

For the general VLM benchmarks, we evaluated Geo-R1, the base model Qwen2.5-VL-7B-Instruct, as well as Geo-R1-Zero and Geo-SFT, to demonstrate this post-training process's ability to preserve the base model's original capabilities.

## L.1. MEGA-Bench

MEGA-Bench is a large-scale multimodal benchmark comprising 8185 manually-annotated examples from 505 tasks. The dataset is designed to cover diverse real-world VLM capabilities across varied input types (images, documents, videos, UI,

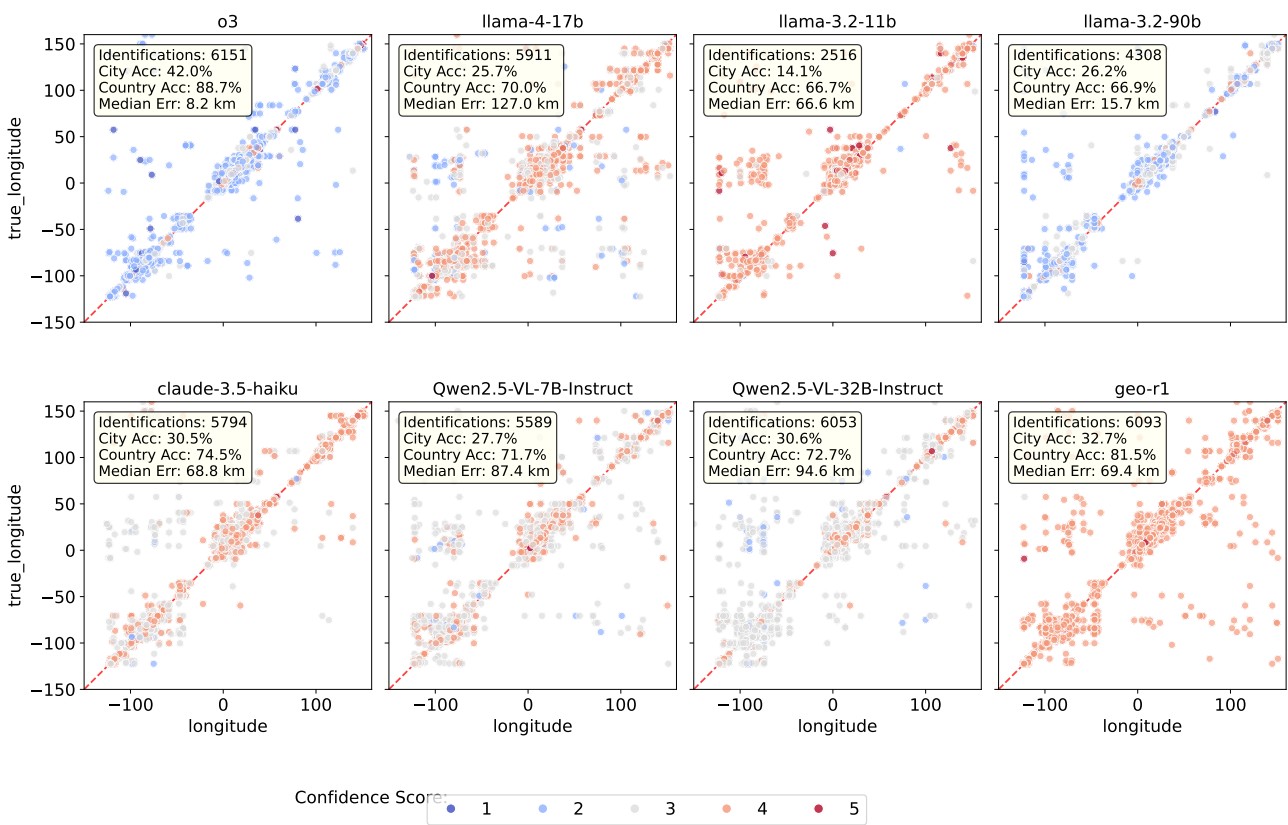

*Figure 23.* Longitutde analysis of IMAGEO-GSS

infographics, etc.) and output formats (text, numbers, LaTeX, code, JSON, structured plans). Instead of relying completely on multiple-choice, it supports rich answer types evaluated with over 45 tailored metrics, combining rule-based checks with LLM-as-judge scoring for open-ended responses.

We evaluted on both the 'core' and 'open' problem sets. For all models we used 512 token max completion length. For the LLM judge, we used GPT-4o, api-version *2025-01-01-preview*. We split the evaluation using the 10 high-level tasks in the benchmark. See Table 15.

*Table 15.* Test Results on Mega-Bench.

| Category | Qwen2.5-VL-7B | Geo-SFT | Geo-R1-Zero | Geo-R1 |
|---|---|---|---|---|
| Commonsense and Social Reasoning | 0.46810 | 0.46181 | 0.49149 | 0.45442 |
| Domain-Specific Knowledge and Skills | 0.36163 | 0.33589 | 0.35176 | 0.34114 |
| Ethical and Safety Reasoning | 0.59229 | 0.58481 | 0.57834 | 0.56829 |
| Language Understanding and Generation | 0.43942 | 0.38218 | 0.44582 | 0.37890 |
| Mathematical and Logical Reasoning | 0.31291 | 0.23013 | 0.31707 | 0.28354 |
| Object Recognition and Classification | 0.37876 | 0.31247 | 0.40089 | 0.34824 |
| Planning and Decision Making | 0.08823 | 0.08763 | 0.15305 | 0.09632 |
| Scene and Event Understanding | 0.39181 | 0.34985 | 0.43067 | 0.38939 |
| Spatial and Temporal Reasoning | 0.24667 | 0.20043 | 0.27925 | 0.26250 |
| Text Recognition (OCR) | 0.46050 | 0.40462 | 0.44696 | 0.36919 |

## L.2. MMMU

The Massive Multi-discipline Multimodal Understanding and Reasoning Benchmark for Expert AGI (MMMU) is a large-scale benchmark of 11.5K multimodal, college-level questions spanning six disciplines, 30 subjects, and 183 subfields, using 30 image types such as diagrams, medical scans, chemical structures, sheet music, and comics. The benchmark

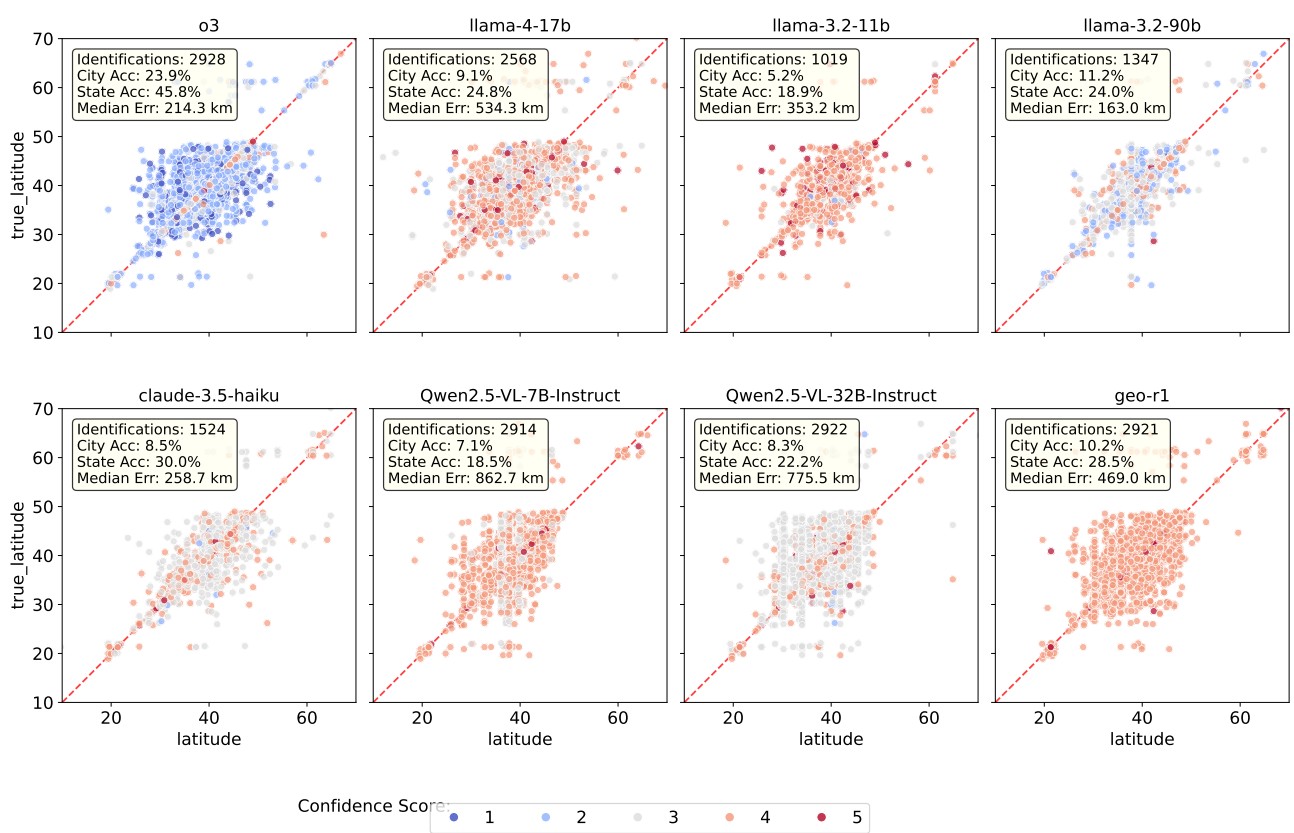

*Figure 24.* Latitude analysis of IMAGEO-UPC

emphasizes both breadth (coverage across many domains) and depth (expert-level reasoning difficulty). Questions, mostly multiple-choice with some open-ended, require models to integrate visual perception, domain-specific knowledge, and deliberate reasoning. We evaluate our models on the MMMU-Val and MMMU-Test sets, with 512 token max completion length. See Table 16.

*Table 16.* Model accuracy on MMMU (Yue et al., 2024) Dev. and Validation splits.

| Model | MMMU Dev. | MMMU Val. |
|---|---|---|
| Qwen2.5-VL-7B-Instruct | 58.0 | 54.2 |
| Geo-SFT | 56.7 | 53.4 |
| Geo-R1-Zero | 50.7 | 54.3 |
| Geo-R1 | 54.0 | 51.2 |

*Table 17.* Model performance on GPQA benchmark results ('extended' dataset).

| Model | Accuracy (%) | Refusal Rate (%) |
|---|---|---|
| Geo-SFT | 31.1 | 1.1 |
| Geo-R1-Zero | 33.0 | 1.5 |
| Geo-R1 | 33.7 | 0.0 |
| Qwen-2.5-VL-7B | 34.2 | 3.3 |

## L.3. GPQA

GPQA is a graduate-level, expert-curated benchmark of multiple-choice questions in physics, chemistry, and biology, designed to be objective and difficult to solve via basic internet search. The dataset is compact but rigorous. Authored and validated by PhD experts, the dataset highlights challenges that lie beyond the reach of non-experts, who achieve 30–34%

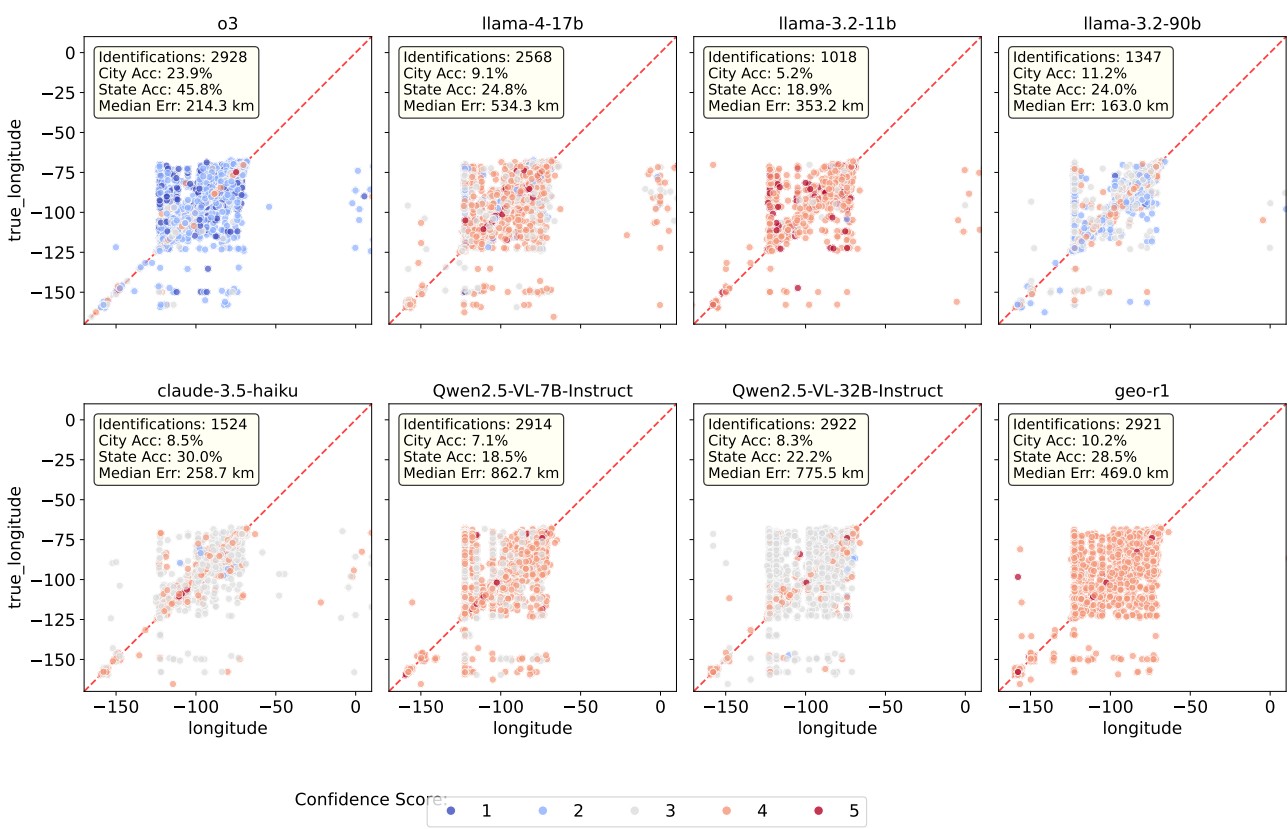

*Figure 25.* Longitutde analysis of IMAGEO-UPC

accuracy even with internet access, compared to experts' 72–81%. We evaluate our post-training checkpoints, editing only lightly the authors evaluation code. We use thed GPQA-Extended dataset, which has 546 questions. See Table 17. For all models we used 1000 token max completion length.

## M. Case Study

We include some case study including a failure case to demo our `Geo-R1` model. All the model using the prompt: "Find the location of the given image." See the details in Fig. 27, 28, and 29.

## N. The Use of Large Language Models

Large Language Models (LLMs), specifically ChatGPT and Gemini, were used as an auxiliary tool in the preparation of this paper. The assistance was limited to polishing writing from a grammatical perspective. No LLMs were used for data generation, experimental results, or research ideation. The authors take full responsibility for all contents of the paper.

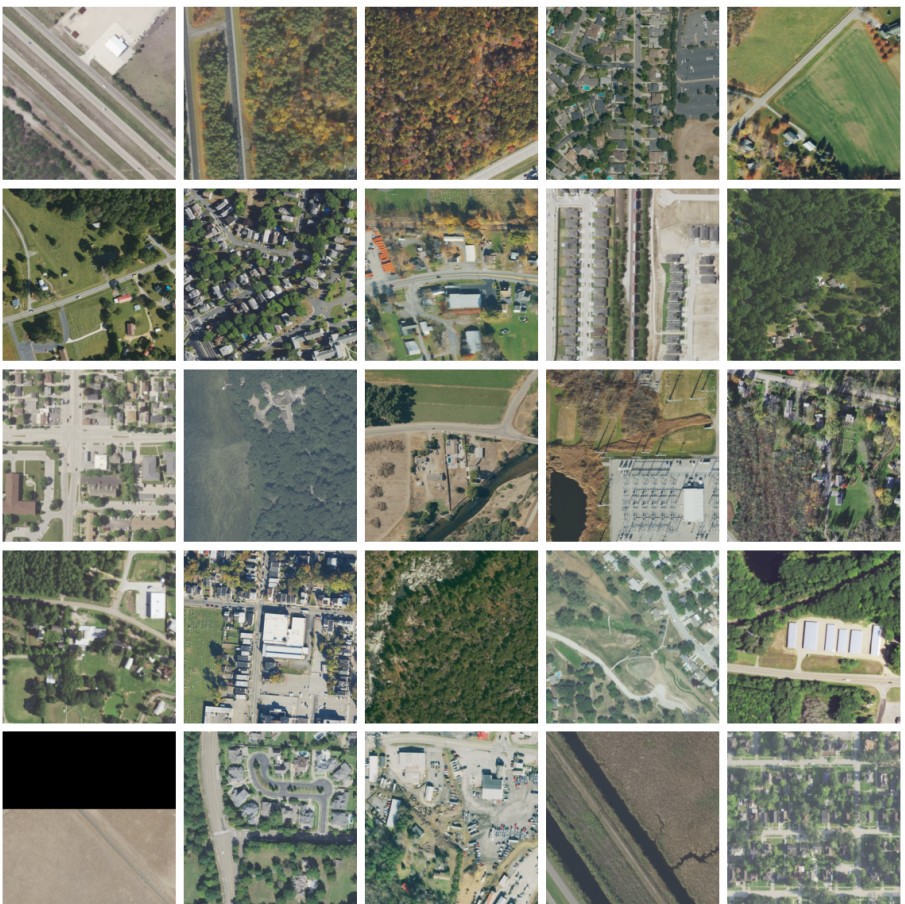

*Figure 26.* Random subset of NAIP images used for aerial image geolocation benchmarking, derived from (Ge et al., 2025).

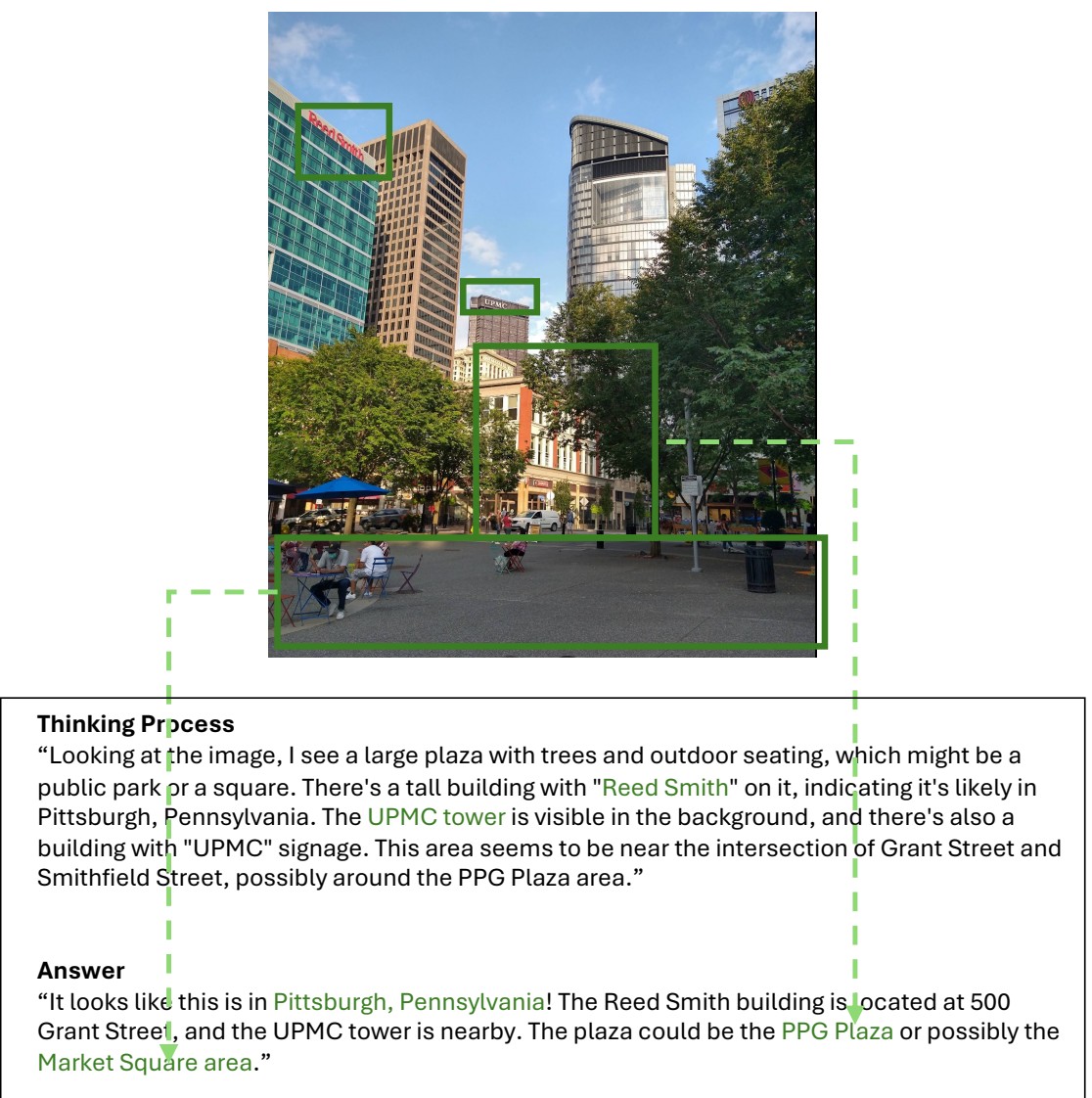

**Thinking Process**
"Looking at the image, I see a large plaza with trees and outdoor seating, which might be a public park or a square. There's a tall building with "Reed Smith" on it, indicating it's likely in Pittsburgh, Pennsylvania. The UPMC tower is visible in the background, and there's also a building with "UPMC" signage. This area seems to be near the intersection of Grant Street and Smithfield Street, possibly around the PPG Plaza area."

**Answer**
"It looks like this is in Pittsburgh, Pennsylvania! The Reed Smith building is located at 500 Grant Street, and the UPMC tower is nearby. The plaza could be the PPG Plaza or possibly the Market Square area."

*Figure 27.* Case study: The `Geo-R1` model can find the visual cues in the image, and link the visual cue to its hidden geospatial knowledge.

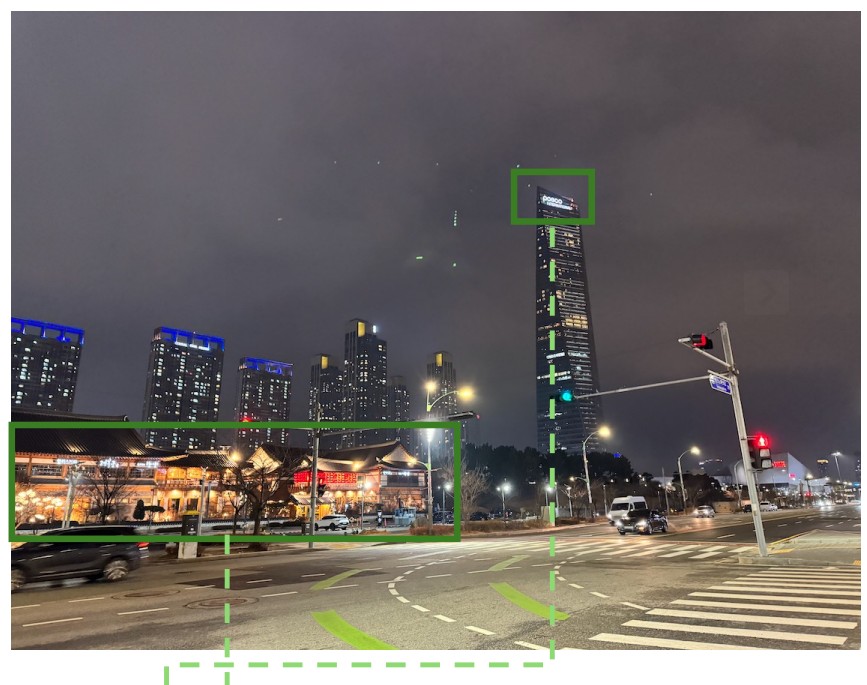

**Thinking Process**

"Alright, I'm analyzing this image. It seems to be a night scene of a city intersection with a tall tower that has "POSCO" written on it. There's also a traditional Korean building nearby, which might be a restaurant or temple. The POSCO tower is located in Songdo International Business District, Incheon, South Korea. I'm recalling that there's a traditional Korean building called "Songdo Hanok Village" near the POSCO tower. I'll check my memory for more details."

**Answer**

"It looks like the POSCO tower is in Songdo, Incheon, South Korea. The traditional Korean building could be "Songdo Hanok Village," which is close to the POSCO tower. So, I think the location is likely Songdo International Business District, Incheon."

*Figure 28.* Case study: The Geo-R1 model can find the visual cues in the image, even in dark environment, where most geolocation models fail. It links the visual cue to its hidden geospatial knowledge.

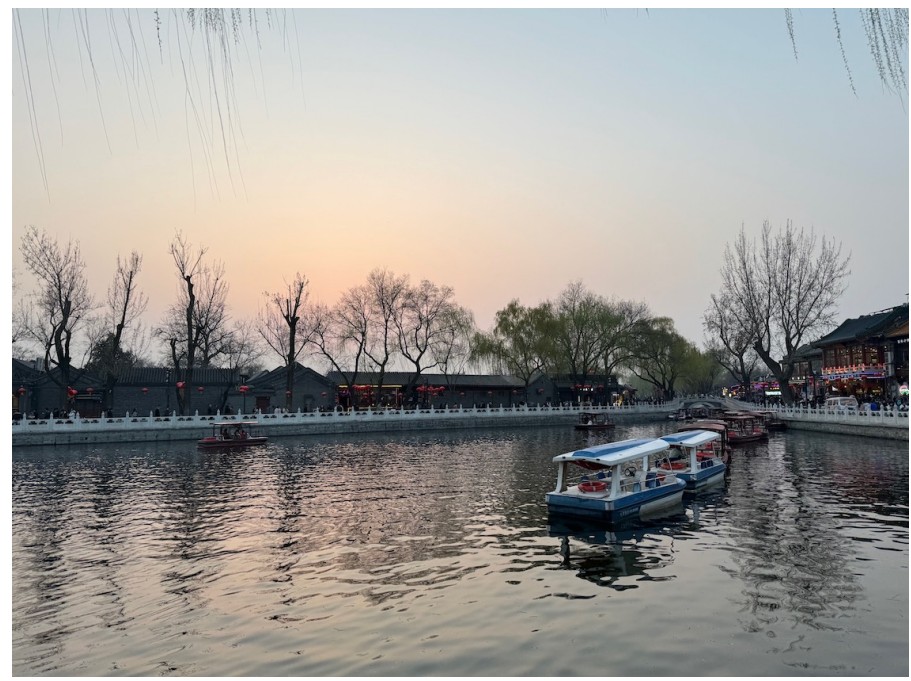

**Thinking Process**

"Looking at the image, I see a canal with boats, possibly rowboats or small motorboats, and a white railing along the bank. There's a cluster of buildings that seem to be traditional Chinese architecture, possibly with red lanterns. The trees appear to be willows, which are common in Beijing. The setting seems to be a tourist area, possibly near the Houhai lakes or the Qianmen district. I'm considering if this could be the area around the Qianmen Bridge or perhaps the Liulichang Street area."

Over Thinking!

**Answer**

"It looks like it could be near the Houhai lakes in Beijing, China. The canal, boats, and traditional Chinese buildings suggest it might be the area around Qianmen or Liulichang. The white railing and red lanterns add to the authentic feel!"

*Figure 29.* Failure case study: The Geo-R1 model may overthink, especially when multiple locations share highly similar geographical features. In this case, it had already provided the correct answer for "Houhai Park, Beijing." However, it overthought and directed the answer to two locations in Beijing that were not precisely the one depicted in the photo..

