# OpenReview forum: "Unlocking Zero-Shot Geospatial Reasoning via Indirect Rewards"
_ICML.cc/2026/Conference — ICML 2026 regular_

### Official Review · Reviewer_Ufwx · 2026-02-25

**Soundness:** 3
**Presentation:** 2
**Significance:** 3
**Originality:** 2
**Overall Recommendation:** 3
**Confidence:** 3

**Summary:**

The paper proposes indirect verifiable rewards based on geolocation metadata via cross-view alignment between ground images and aerial images for scalable geospatial reasoning capability. The framework Geo-R1 consists of 2 stages: CoT-SFT and GRPO-based post-training. The CoT-SFT stage initializes the model with the thinking paradigm to reason about geography with sythesized CoT data, followed by GRPO  with proposed indirect verifiable rewards. The experiments demonstrate strong zero-shot performance on downstream OOD tasks.

**Compliance With Llm Reviewing Policy:**

Affirmed.

**Final Justification:**

Thank authors for detailed response. However, consider potential major revision of the manuscript, I will maintain my score.

**Key Questions For Authors:**

1. For indirect verifiable reward, it's defined as binary {-1, 1} reward according to the main paper but {-1, -0.8, 1} in the appendices. The definition itself is contradictory, clarification is needed.
2. The hyperparameter k (candidate set size) is not specified. Authors should provide detailed analysis for the choice as it's the main design component. The policy for choosing hard negatives should be clearly stated, rather than simply mentioning 'same spatiotemporal neighborhood' in the paper. The hypothesis about non-discriminative nuisance factors should be justified and more explanations are needed for readers to better understand the concept introduced.
3. The theorem is hard to follow and seems redudant to the proposed method, where pair-consistent alignemnt between ground-satellite imagery is a common strategy and empirically validated in many works, i.e.,GRAFT [1].
4. The presentation is not straightforward enough, it can be better formulated for easy understanding. Also, the methodology is not clearly presented in the main pages.
5. I wonder the performance of a supervised finetuned model with the same cross-view paring task (no RL) to show the impact of RLVR.

[1] GRAFT: REMOTE SENSING VISION-LANGUAGE FOUNDATION MODELS WITHOUT ANNOTATIONS VIA GROUND REMOTE ALIGNMENT

**Limitations:**

Discussions of limitations and potential societal impact could be helpful.

**Strengths And Weaknesses:**

**Strengths**:
1. The paper is technically sound, the motivation of introducing indirect verifiable rewards into RL post-training for scalable geospatial applications using cross-view pairing with metadata is reasonable, offering a generalizable and cost-effective solution.
2. The method has some significance in geospatial analysis. Zero-shot OOD evaluation across various tasks is compelling and the experiment results in both in-distribution and out-distribution tasks are strong and promising.

**Weaknesses**:

See key questions for details.

---

> ### Author Rebuttal · Authors · 2026-03-31
>
> We sincerely appreciate the constructive comments from reviewer Ufwx and the time spent on reviewing this paper. We address the questions and clarify the issues accordingly as described below.
>
> >**[Q1]** For indirect verifiable reward, it's defined as binary {-1, 1} reward according to the main paper but {-1, -0.8, 1} in the appendices.
>
> **[Re Q1]**: We thank the reviewer for their careful reading and for pointing out this detail. To clarify, in our theoretical analysis and proofs, we formulated the verifiable reward as a strict binary {-1, 1} purely for mathematical simplicity and conceptual clarity. The fundamental geospatial objective *is* conceptually binary: the model either correctly identifies the matching satellite view or it does not.
>
> The {-1, -0.8, 1} values detailed in the appendix reflect the practical engineering implementation designed to penalize format failures without altering this core optimization task. In our framework, we penalize unparsable or malformed outputs with -0.2. For correctly formatted outputs, the accuracy reward is mapped to {-0.8, 1}, resulting {-1,-0.8,1}.
>
> Because GRPO computes **advantages** by normalizing rewards within a sampled group ($A=\frac{r-\mu}{\sigma}$), the {-0.8, 1} implementation is simply an affine transformation ($r_{impl}=0.9\cdot r_{bin}+0.1$) of the pure {-1, 1} binary reward.
>
> Mathematically, both the shift and the scaling factor completely cancel out during the normalization step:
> $$A_{impl}=\frac{(0.9\cdot r_{bin}+0.1)-(0.9\cdot \mu_{bin}+0.1)}{0.9\cdot \sigma_{bin}}=\frac{r_{bin}-\mu_{bin}}{\sigma_{bin}}=A_{bin}$$
> Therefore, the optimization gradients for the cross-view pairing task are mathematically identical to those of a pure binary reward. This formulation allows us to effectively penalize formatting errors without distorting the underlying binary reasoning signal.
>
> >**[Q2]** selection of k, negative policy and nuisance.
>
> **[Re Q2}**: We use k=5 (1 target, 4 confusers). This balances memory constraints with task difficulty. Random guessing yields 20% (our zero-shot baseline gets 19%), providing a sufficient "entropy gap" for RL optimization without exceeding GPU memory limits when processing multiple high-res images. We are actively training on other k like 4,7,10 for ablation, we will update it one the training finished
>
> **Hard Negative Policy:** Confusers are sampled strictly from the *same city* as the target (via CV-Cities) with distinct coordinates. This ensures they share regional visual traits but have strictly non-overlapping satellite fields-of-view.
>
> **Nuisance factors** (N) are superficial cues like weather, lighting, or regional architecture. Since all k candidates are from the same city, these traits are uniform across the set. Thus, the mutual information between the target and these factors is close zero ($I(Y; N) \to 0$). This prevents shortcut learning, theoretically forcing the model to rely solely on fine-grained, view-invariant geometric structures (e.g., road topologies) to solve the task.
>
> >**[Q3]** The theorem ...
>
> **[Re Q3]**: We agree cross-view alignment is an established, effective strategy for representation learning and will prominently cite it. However, our theorem serves a fundamentally different purpose. While GRAFT uses alignment as a contrastive objective for latent embeddings, Geo-R1 employs it as an environment with verifiable proxy rewards for RLVR to elicit explicit CoT reasoning.
>
> Our theorem is not intended to prove that pairing is a good visual task, which is empirically known. Instead, it provides a theoretical guarantee for why RLVR elicits generalizable reasoning from a simple binary reward. It mathematically proves that by factoring out nuisance variables (e.g., weather, regional styles), optimizing a {-1, 1} reward strictly forces the model to learn view-invariant geometric logic, preventing superficial shortcut learning. This theoretical link between indirect proxy rewards and the zero-shot emergence of structural reasoning is a unique contribution not covered in empirical literature. We will add a preamble to explicitly clarify this distinction.
>
>
> >**[Q4]** The presentation is not straightforward enough...
>
> **[Re Q4]**: The strict page limit forced us to condense the methodology and defer critical details (Appendix A,B,C) and pipeline figures （Fig.11）to the appendix. To address this, we will fully utilize the extra page in the camera-ready version to move these detailed formulations and system figures back into the main text. This restructuring will provide a clear, self-contained visual walkthrough of our framework, significantly improving the methodology's clarity and flow.
>
> >**[Q5]** SFT model with the same cross-view paring task.
>
> **[Re Q5]**: We have included experimental results of the same dataset using SFT and DPO. Results show that our method demonstrates better performance and generalization compared to SFT and DPO. For details, please refer to our response to reviewer wFD1 W2.

---

> > ### Author Rebuttal · Reviewer_Ufwx · 2026-04-03
> >
> > Thank you for your reply. I have a few remaining concerns regarding the reward design and overall presentation.
> >
> > 1. Since the overall reward formulation already includes a format reward, it is unclear why an additional malformed penalty of −0.2 is introduced within the accuracy reward.
> >
> > 2. I am not fully convinced by the motivation for introducing the theorem. It is now widely recognized that RLVR can elicit generalizable reasoning even from simple binary reward signals, which makes the added theoretical component feel less essential without stronger justification. The presentation needs substantial refinement.

---

> > > ### Author Response · Authors · 2026-04-03
> > >
> > > We sincerely thank the reviewer for their feedback. In response to the two follow-up questions raised, we provide the following answers, which we hope will address the outstanding questions posed by the reviewer.
> > >
> > > >**[FQ1]** Since the overall reward formulation already includes a format reward, it is unclear why an additional malformed penalty of −0.2 is introduced within the accuracy reward.
> > >
> > > **[Re FQ1]** That’s an excellent question, and we’re very grateful to the reviewer for raising it. The additional penalty of -0.2 and the format reward serve as two entirely distinct evaluation metrics.
> > > Our cross-view pairing defines the task as selecting the correct satellite image from multiple options, each labeled with an index (A–E). This additional -0.2 penalty is applied when the final output exceeds the specified index range. The format reward, on the other hand, evaluates whether the VLM can generate responses in the format `<think>`...`<\think>` `<answer>`…`<\answer>`. Let’s take an example and assume that satellite image A is the correct answer.
> > >
> > > - When the model outputs "  `<think>`Reason1, Reason2, …`<\think>` `<answer>`A`<\answer>` ", it will receive an accuracy reward of 1 and a format reward of 1.
> > >
> > > - When the model outputs "  `<think>`Reason1, Reason2, …`<\think>` `<answer>`C`<\answer>` ", it receives an accuracy reward of -0.8 and a format reward of 1.
> > > - When the model outputs "  `<think>`Reason1, Reason2, …`<\think>` `<answer>`Whatever beyond A-E`<\answer>` ", it receives an accuracy reward of -1 and a format reward of 1.
> > > - When the model outputs " Reason1, Reason2 `<answer>`A`<\answer>` ", it receives an accuracy reward of 1 and a format reward of 0.
> > > - When the model outputs " Reason1, Reason2 `<answer>`C`<\answer>` ", it receives an accuracy reward of -0.8 and a format reward of 0.
> > > - When the          `<answer>`...`<\answer>` box does not even exist, it receives an accuracy reward of -1 and a format reward of 0.
> > >
> > > In other words, this -0.2 reward can be interpreted as a reward for the model correctly recognizing that it is solving a multiple-choice problem. The format reward, on the other hand, is a reward for the model’s ability to perform implicit CoT.This refined design allows for a more direct and stable RL training process. Of course, we could split this 0.2 reward into a separate “Multi-Choice Reward.” Our new analysis shows that, as the model trains, the trend of the curve for this multi-choice reward does not align with that of the format reward. We will include a separate discussion of this aspect in the next version.
> > >
> > > >**[FQ2]** I am not fully convinced by the motivation for introducing the theorem. It is now widely recognized that RLVR can elicit generalizable reasoning even from simple binary reward signals, which makes the added theoretical component feel less essential without stronger justification. The presentation needs substantial refinement.
> > >
> > >
> > > **[Re FQ2]** We sincerely thank you for the continued dialogue and for pushing us to improve the paper's readability. We agree that the current presentation of the theoretical analysis disrupts the narrative flow.
> > >
> > > We agree that RLVR’s effectiveness with binary rewards is well-recognized. Building on this consensus, our theoretical analysis addresses the unique challenge of "shortcut learning" in continuous visual domains, where models often match superficial textures rather than developing genuine structural reasoning. Rather than serving as a rigid mathematical proof, this section acts as a conceptual framework. It illustrates how our task design structurally mitigates these visual shortcuts by factoring out nuisance variables, fostering the emergence of robust spatial logic.
> > >
> > > That said, we **completely agree** that the dense mathematical formulation is not necessary for the main text. To substantially refine the manuscript based on your feedback, we will implement the following steps：
> > >
> > > 1. Refining into a Conceptual Framework: We will **move the detailed analytical derivations and equations entirely to the appendix**. In the main text, we will reframe the "Theorem" into a brief, intuitive "Theoretical Framework" that highlights the Vision vs. Symbolic RLVR distinction and explains how **our design philosophy** structurally avoids visual shortcuts.
> > >
> > > 2. Promoting the Methodology: By removing the theoretical bloat, we free up critical space in the main pages. We will utilize this space to **bring the core methodology details**, step-by-step formulations, and the visual pipeline figures (currently relegated to the appendix) back into the main text.
> > >
> > > This restructuring effectively solves both presentation issues: it replaces the heavy theoretical complexity with an accessible conceptual framework, and provides a straightforward, visually intuitive formulation of our methodology right in the main pages.

---

### Official Review · Reviewer_jZiA · 2026-03-12

**Soundness:** 2
**Presentation:** 3
**Significance:** 3
**Originality:** 2
**Overall Recommendation:** 5
**Confidence:** 4

**Summary:**

This manuscript addresses the challenge of training reasoning-capable VLMs in the geospatial domain, where raw imagery is abundant but high-quality reasoning annotations are scarce.
The authors introduce Geo-R1, a two-stage post-training framework that utilizes indirect, verifiable rewards derived from geolocation metadata to induce zero-shot geospatial reasoning.
Stage 1 involves "Scaffolding," using a small synthetic dataset of 12.6k reasoning trajectories to instill a thinking paradigm.
Stage 2, "Reasoning Elevation," employs GRPO on a "Cross-View Pairing" proxy task, where the model must match ground-level panoramas to corresponding satellite images among hard negatives.
Evaluated on downstream tasks, Geo-R1 shows significant improvements in zero-shot generalization on benchmarks like GeoChain and IMAGEO-Bench, even surpassing specialized models while maintaining general-purpose VLM capabilities.

**Compliance With Llm Reviewing Policy:**

Affirmed.

**Final Justification:**

As most of my concerns have been addressed, I choose to raise my score.
I encourage the authors to further improve the manuscript by incorporating the feedback provided during the rebuttal.

**Key Questions For Authors:**

Please see the Weaknesses. I will consider raising the score based on the final rebuttal results.

Minor Formatting Issue: There is a double quote error around lines 182-183.

**Limitations:**

The authors have included an Impact Statement addressing societal needs and potential benefits like disaster response.
They also provide a failure case study in Appendix.
However, as noted in the Weaknesses, they have not adequately discussed the specific limitations of the Geo-R1 framework within the main text.
I suggest adding a dedicated section in the main paper that synthesizes the failure modes (like "overthinking" in Fig. 29)

**Strengths And Weaknesses:**

Strengths:
- The manuscript is exceptionally well-written, providing a balanced mix of theoretical proofs and experimental analysis that makes the proposed framework accessible
- To validate the effectiveness of Geo-R1, the authors tested the model against a massive array of benchmarks, all of which show robust performance gains
- The model shows a remarkable capacity for zero-shot transfer to out-of-distribution tasks, without ever seeing direct supervision for those tasks during the elevation phase
- The appendix contains a wealth of technical details, training dynamics, and additional experimental results that bolster the reproducibility and depth of the study
- The authors successfully demonstrate that specialized geospatial training does not lead to catastrophic forgetting of base model skills, as evidenced by stable performance on MMMU, MEGA-Bench and GPQA

Weaknesses:
- While the application is successful, the use of GRPO-style reinforcement learning in the image geolocalization and geospatial domain is not entirely new, which slightly reduces the overall methodological novelty of the submission
- Although there is a failure case study provided in Fig. 29 within the appendix, the main body of the paper lacks a dedicated, thorough discussion regarding the method’s limitations or potential failure modes. Discussing these constraints in the core text is vital for a balanced academic assessment
- The paper would benefit from a more granular ablation study (e.g., reward components or negative sampling)

---

> ### Author Rebuttal · Authors · 2026-03-31
>
> We sincerely appreciate the constructive comments from reviewer jZiA and the time spent on reviewing this paper. We address the questions and clarify the issues accordingly as described below.
>
> >**[W1]** Methodological novelty.
>
> **[Re W1]**:  While we leverage established techniques like RL, our core theoretical and methodological novelty lies in formulating cross-view metadata as an indirect, verifiable reward for complex reasoning. Although previous work like GLOBE[1] have pioneered RL in specific geo-localization task, they heavily rely on task-direct strong rewards (dense human annotations) for in-distribution performance gains. Training reasoning models in specialized domains like remote sensing is bottlenecked by the extreme scarcity of dense human annotations, and as a result, it is hard to scale. By both mathematically and empirically demonstrating that solving the proxy matching task forces the internalization of transferable physical laws, we provide a highly scalable, novel paradigm for domain-specific VLM post-training without requiring human-annotated reasoning traces.
>
> [1] Ling Li, et al. "Recognition through Reasoning: Reinforcing Image Geo-localization with Large Vision-Language Models". NeurIPS 2025
>
> >**[W2 & Limitations]** Failure analysis and limitations.
>
> **[Re W2 & Limitations]**: We sincerely thank the reviewer for this constructive suggestion. We completely agree that a transparent discussion of the method's constraints is vital for a balanced academic assessment. In the camera-ready version, we will add a dedicated "Limitations and Failure Modes" section directly into the main body of the paper and provide further failure case analysis in appendix (expanding upon the case study currently in Appendix Fig. 29).
>
> This new section will provide a thorough qualitative analysis of where the indirect verifiable reward encounters physical or algorithmic bottlenecks:
>
> 1. **Texture Uniformity & Scale Ambiguity:** While our framework excels in structured environments, it struggles in highly featureless landscapes (e.g., dense uniform forest canopies, open water, or vast deserts). In these areas, the lack of distinct structural landmarks prevents the cross-view geometric anchoring mechanism from establishing reliable reference points, causing the model's spatial reasoning to degenerate into random guessing.
>
> 2. **Severe Illumination & Temporal Shifts:** The framework is inherently bottlenecked by the robustness of the base visual encoder under extreme domain gaps. When matching daytime satellite imagery with extreme night-time street views or heavily snow-covered scenes, the visual encoder occasionally drops critical low-level geometric cues (e.g., road lane markers). Without these foundational visual tokens, the RL-trained reasoning module cannot successfully execute the learned cross-view alignment.
>
> 3. **Physical Occlusions:** Geo-R1 naturally encounters difficulties with ground-level images taken inside deep urban canyons, under dense foliage, or near indoor-outdoor boundaries. In these scenarios, the spatial structures visible from the ground are physically occluded from the overhead satellite perspective, breaking the fundamental assumption of our cross-view indirect reward mapping.
>
> >**[W3]** Granular ablation.
>
> **[Re W3]**
> We will add these ablations to the camera-ready version, as well as k's impact. Additionally, we have included results using the same pairing dataset on SFT and DPO to demonstrate the effectiveness of GRPO. For details, please refer to the response to reviewer wFD1 W2.
>
> **Reward Components:**
> The reward components we employ, including format, length, and repetition rewards, are widely accepted by the community as effective methods for stabilizing the training process and have been extensively validated. While our computational resources are insufficient to support ablation experiments for each reward individually, we conducted a comparison between an “accuracy-only” approach, an “accuracy-excluded” approach, and the full reward setup. As shown in the table, this comparison fully demonstrates the effectiveness of the indirect reward we propose as the accuracy reward signal.
>
> Reward| Format Adherence | ID Acc | OOD City Acc
> -|-|-|-
> Acc-Only | 34.2% | 41.5% | 18.3%
> Acc-Excluded | 98.2% | 20.5% | 24.3%
> **Full**| **98.7%**| **82.0%**| **32.7%**
>
> **Negative Sampling (Hard vs. Random):**
> With random negatives, the model takes "shortcuts" relying on superficial nuisance factors (e.g., climate, vegetation color), failing on OOD tasks(same as baseline). Hard negatives share identical styles, making nuisance factors non-discriminative. The model is mathematically forced to learn view-invariant geometric alignments (e.g., road intersections), preventing shortcut learning.
>
> Negative Sampling|ID Acc|OOD City Acc
> -|-|-
> Random(Cross-city) | **83.4%** | 27.1%
> **Hard (same city - Ours)**| 82.0% | **32.7%**
>
> >**[Minor Formatting Issue]** We will fix it in the next release.

---

> > ### Author Rebuttal · Reviewer_jZiA · 2026-04-03
> >
> > I acknowledge the authors’ clarification on methodological novelty. Nonetheless, it should be noted that GLOBE does not involve human-annotated reasoning traces. Despite this, I find the perspective of the proposed idea to be novel, and the additional experiments have addressed my concerns.
> > Based on this, I will raise my score accordingly.

---

> > > ### Author Response · Authors · 2026-04-08
> > >
> > > We sincerely thank the reviewers for their support in accepting our paper. We will carefully consider the reviewers’ comments and incorporate these valuable suggestions into the next release of our paper.

---

### Official Review · Reviewer_v1J7 · 2026-03-12

**Soundness:** 2
**Presentation:** 3
**Significance:** 2
**Originality:** 2
**Overall Recommendation:** 3
**Confidence:** 3

**Summary:**

This paper investigates whether large vision–language models (VLMs) can acquire geospatial reasoning ability through post-training using indirect supervision derived from cross-view alignment metadata. The proposed approach trains a VLM using satellite–street view correspondences and metadata-based rewards that enable verification of geospatial reasoning without requiring dense human annotations. The model is post-trained using this signal to encourage reasoning about spatial relationships between views.

The resulting model is evaluated on a wide range of benchmarks, including GeoChain, IMAGEO-GSS, MP16-Reason, and GeoBench-VLM. The authors report consistent improvements over the base Qwen2.5-VL-7B model and show that the model transfers to several downstream benchmarks without additional task-specific training. The paper further argues that the training procedure enables zero-shot geospatial reasoning while avoiding catastrophic forgetting of the base model’s capabilities.

Overall, the paper presents a comprehensive empirical study exploring how indirect geospatial supervision can improve cross-view understanding and reasoning capabilities in multimodal models.

**Compliance With Llm Reviewing Policy:**

Affirmed.

**Key Questions For Authors:**

***Geospatial reasoning vs general reasoning***: Some GeoBench tasks (e.g., aircraft counting and vehicle counting) appear to require structured visual reasoning rather than geospatial reasoning specifically. How do the authors distinguish improvements arising from geospatial training signals versus improvements due to the model becoming a stronger general reasoning model after RL-based post-training?

***Effect of reasoning prompting***: Have the authors evaluated the base model using explicit reasoning prompts (e.g., prompting the model to reason step-by-step before answering)? In related benchmarks, we have observed that such prompting can recover a substantial portion of the performance gains relative to the base model. Such a comparison could help disentangle the effect of RL training from improvements due to reasoning-style prompting.

***Definition of OOD generalization***: In several experiments, OOD generalization appears to correspond to unseen cities. Could the authors clarify how distribution shift is defined in these settings and whether stronger shifts (e.g., different geographic regions or environments) were considered?

***Catastrophic forgetting***: The paper claims that the approach avoids catastrophic forgetting, yet Appendix L appears to show consistent performance degradation on some primitive tasks. Could the authors clarify how catastrophic forgetting is defined in this context and how these degradations should be interpreted?

**Limitations:**

No. The paper does not appear to include a dedicated discussion of limitations. It would be helpful for the authors to include a short section outlining the limitations of the proposed approach, such as the extent to which the method truly learns geospatial reasoning versus general reasoning behaviors induced by RL-based post-training, the relatively limited definition of OOD generalization used in the experiments (e.g., unseen cities), and potential degradation of base model capabilities observed in Appendix L.

**Strengths And Weaknesses:**

# Soundness

### Strengths
- The training approach using verifiable metadata-based rewards derived from cross-view alignment is technically well motivated and aligns with recent work on reinforcement learning with verifiable rewards for reasoning tasks.
- The paper provides extensive empirical evaluation across multiple datasets and benchmarks, including ablations and comparisons with several baselines.
- The improvements over the base model appear consistent across many tasks, suggesting that the training signal is effective.

### Weaknesses
- It is unclear whether the improvements arise from learning geospatial reasoning specifically or from the model becoming a stronger general reasoning model after RL-based post-training. For example, some GeoBench tasks (e.g., aircraft or vehicle counting) require structured visual reasoning rather than geospatial reasoning. Without additional controls, it is difficult to disentangle whether the gains stem from geospatial supervision or from the RL procedure encouraging reasoning-style inference.
- Some claims appear stronger than what the experiments directly support. In particular, the framing around zero-shot geospatial reasoning and OOD generalization seems broader than what is demonstrated, as many evaluated tasks remain closely related to geolocalization or cross-view matching. In several cases, the OOD setting corresponds to unseen cities, which may still share strong distributional similarities with the training data.
- The paper claims that the approach avoids catastrophic forgetting, yet Appendix L appears to show consistent performance degradation on some primitive tasks. While the degradation may be modest, this seems somewhat inconsistent with the stronger claim made in the main text.

---

# Presentation

### Strengths
- The paper is generally well organized, with a clear description of the training setup and a comprehensive experimental section.
- The inclusion of multiple benchmarks and ablations helps readers understand the breadth of evaluation.

### Weaknesses
- Some terminology is used somewhat broadly, particularly the notion of “geospatial reasoning,” which appears to encompass both reasoning tasks and standard geolocalization benchmarks.
- The paper would benefit from deeper qualitative analysis of how the model performs geospatial reasoning, including failure mode analysis.

---

# Significance

### Strengths
- The idea of leveraging metadata-derived supervision to train reasoning capabilities without manual annotation is promising and may generalize to other domains.

### Weaknesses
- Although the evaluation includes tasks beyond cross-view localization (e.g., GeoBench tasks such as vehicle or aircraft counting), it is unclear whether improvements on these tasks reflect geospatial reasoning ability or improvements in general reasoning capabilities induced by RL-based post-training.
- As a result, it remains difficult to isolate how much of the observed performance gains stem specifically from geospatial supervision, as opposed to the model becoming a stronger general reasoning model. This makes the broader significance of the method for advancing geospatial reasoning somewhat unclear.

---

# Originality

### Strengths
- The combination of cross-view alignment signals with reasoning-style post-training for VLMs is an interesting and relatively unexplored direction.
- The work provides a large-scale empirical exploration of this idea across many benchmarks.

### Weaknesses
- Many of the individual components (cross-view alignment supervision, RL-based reasoning post-training) build on existing ideas. The novelty lies primarily in the integration and empirical demonstration rather than in a fundamentally new modeling approach.

---

> ### Author Rebuttal · Authors · 2026-03-29
>
> We sincerely appreciate the constructive comments from reviewer v1J7 and the time spent on reviewing this paper. We address the questions and clarify the issues accordingly as described below.
>
> >**[KQ1]** Geospatial reasoning vs general reasoning tasks.
>
> **[Re KQ1]**: Regarding the observation that tasks like aircraft or vehicle counting in GeoBench appear to require standard "structured visual reasoning," we respectfully clarify that interpreting remote sensing images is inherently a geospatial problem. Unlike ground-level images, satellite imagery presents top-down silhouettes, and the pixel size of objects is strictly governed by spatial resolution (Ground Sample Distance) and altitude. **Generic visual reasoners struggle here.** Our indirect reward explicitly forces the model to map these scale variations and cross-view geometric priors into its weights, providing the necessary geospatial foundation for these specific tasks.
>
> >**[KQ2]** Effect of reasoning prompting.
>
> **[Re KQ2]**: To empirically prove that our gains come from these learned geospatial priors rather than just general "reasoning-style inference", we conducted a full ablation study on the IMAGEO-GSS benchmark:
>
> model|thinking prompt|city_acc|country_acc
> -|-|-|-
> base|❌|27.73%|71.65%
> -|✅|27.80%|71.15%
> Geo-R1|❌|30.12%|78.44%
> -|✅|32.73%|81.46%
>
> This table clearly demonstrates two critical points:
> 1. **General reasoning alone is insufficient:** Explicitly prompting the base model to reason only marginally improves its performance on city-level localization (27.73% -> 27.80%), indicating a severe lack of spatial priors.
> 2. **Priors are embedded in the weights:** Most importantly, even when Geo-R1 operates *without* a thinking prompt (acting purely on its learned implicit priors), it significantly outperforms the base model operating *with* a thinking prompt (City Acc: 30.12% vs. 27.80%). This conclusively demonstrates that our RL framework successfully internalizes robust, domain-specific spatial mapping.
>
> >**[KQ3 & W2]** Definition of OOD generalization and zero-shot claims.
>
> **[Re]**: First, our zero-shot claims emphasize **task-level generalization**, as the evaluation benchmarks use entirely different task formulations, camera parameters, and sampling altitudes compared to our training set.
>
> Second, regarding geographic distribution shift, there are **strictly zero overlapping image samples** between training and evaluation. As detailed in our data **table in response to Reviewer wFD1**, 92.07% of the IMAGEO-GSS evaluation dataset consists of strictly unseen cities. Crucially, Geo-R1 achieves a **+5.1% absolute accuracy gain on these strictly non-overlapping cities**. This refutes the hypothesis that the model relies on distributional similarity, proving true zero-shot transfer to novel geographic environments.
>
> >**[KQ4 & W3]** Definition of catastrophic forgetting.
>
> **[Re KQ4 & W3]**: In standard domain-specific full fine-tuning, general-purpose capabilities often suffer severe degradation (e.g., an MMMU score collapsing from 0.6 to 0.2), which is known as catastrophic forgetting. In contrast, the performance fluctuations observed in Geo-R1 on general benchmarks are extremely minor. When weighed against the massive gains achieved in complex geospatial reasoning, this marginal trade-off represents a highly favorable outcome. We will revise the text to explicitly clarify this definition.
>
> >**[Presentation W1, W2, Lim]** Terminology and Qualitative Analysis and limitations.
>
> **[Re]**: We will tighten our terminology in the camera-ready version, explicitly distinguishing between "geospatial matching/localization" and "structured geospatial reasoning." We will also add a qualitative failure mode analysis to the appendix, highlighting where the indirect reward struggles:
> 1. **Texture Uniformity & Scale Ambiguity:** In highly featureless environments (e.g., dense uniform forest canopies or open water), the cross-view geometric anchoring mechanism struggles without distinct structural landmarks.
> 2. **Severe Illumination/Temporal Shifts:** When matching daytime satellite imagery with extreme night-time street views, the base visual encoder occasionally drops critical geometric cues before the reasoning module can process them.
>
> >**[Originality W1]** Novelty lies in integration rather than a fundamental new modeling approach.
>
> **[Re]**: While we leverage established techniques like RL and cross-view alignment, our core theoretical and methodological novelty lies in **formulating cross-view metadata as an indirect, verifiable reward for complex reasoning**. Training reasoning models in specialized domains like remote sensing is bottlenecked by the extreme scarcity of dense human annotations. By mathematically demonstrating that solving the proxy matching task forces the internalization of transferable physical laws, we provide a highly scalable, novel paradigm for domain-specific VLM post-training without requiring human-annotated reasoning traces.

---

> > ### Author Rebuttal · Reviewer_v1J7 · 2026-04-04
> >
> > Thanks for the detailed rebuttal. After careful consideration, I maintain my score.
> > Regarding KQ1/KQ2: the ablation presented (base vs. Geo-R1 with/without thinking prompt on IMAGEO-GSS) does not address the original concern. IMAGEO-GSS geolocation is strongly in-distribution for Geo-R1's training objective. A more convincing ablation would test reasoning prompting on the GeoBench tasks (e.g., aircraft/vehicle counting) that originally motivated the question. The claim that geospatial-specific priors — rather than general RL-induced reasoning behaviors — drive the gains on those tasks remains unsubstantiated without a domain-agnostic RL-trained reasoning baseline.
> > Regarding catastrophic forgetting: the rebuttal redefines the threshold for "catastrophic" forgetting rather than addressing the empirical pattern. Tables 15–16 show consistent, broad-based performance declines across nearly all general capability categories (language, math, OCR, MMMU), which is difficult to reconcile with the claim of no catastrophic forgetting, regardless of magnitude.
> > I appreciate the authors' effort in the rebuttal, but these are substantive concerns that would require additional experiments to resolve rather than clarification alone.

---

> > > ### Author Response · Authors · 2026-04-07
> > >
> > > We are very grateful to the reviewers for their engagement in the follow-up discussions. We provide the following additional responses to the two questions raised in the follow-up
> > >
> > > >**[Follow-up Question1]** Ablation Study on GeoBench.
> > >
> > > | Task Category | Qwen2.5-VL-7B (Base) | Qwen2.5-VL-7B (w/ Thinking) | VLM-R1-7B (General RL) | Geo-R1 (w/o Thinking) | Geo-R1 (w/ Thinking) |
> > > | :--- | :--- | :--- | :--- | :--- | :--- |
> > > | **Aircraft Classification** | 0.47 | 0.49 | 0.50 | 0.68 | **0.71** |
> > > | **Building Counting** | 0.14 | 0.17 | 0.22 | 0.43 | **0.49** |
> > > | **Crop Classification** | 0.13 | 0.13 | 0.14 | 0.17 | **0.18** |
> > > | **Disaster Classification** | 0.65 | 0.65 | **0.66** | 0.51 | 0.53 |
> > > | **Fire Risk Assessment** | 0.12 | 0.13 | 0.13 | 0.20 | **0.22** |
> > > | **General Aircraft Counting** | 0.28 | 0.31 | 0.35 | 0.45 | **0.50** |
> > > | **General Vehicle Counting** | 0.29 | 0.32 | 0.36 | 0.47 | **0.50** |
> > > | **Land Use Classification** | 0.57 | **0.58** | **0.58** | 0.57 | **0.58** |
> > > | **Marine Debris Counting** | 0.31 | 0.33 | 0.37 | 0.52 | **0.56** |
> > > | **Scene Classification** | 0.80 | 0.81 | 0.81 | 0.83 | **0.83** |
> > > | **Ship Classification** | 0.39 | 0.41 | 0.42 | 0.47 | **0.49** |
> > > | **Spatial Relationship** | 0.62 | 0.64 | 0.67 | 0.72 | **0.75** |
> > > | **Specific Aircraft Counting**| 0.17 | 0.20 | 0.23 | 0.44 | **0.48** |
> > > | **Specific Vehicle Counting** | 0.23 | 0.26 | 0.29 | 0.49 | **0.52** |
> > > | **Tree Health Assessment** | 0.15 | 0.17 | 0.18 | 0.32 | **0.34** |
> > > | **Trees Counting** | 0.25 | 0.27 | 0.31 | 0.39 | **0.42** |
> > > | **Water Bodies Counting** | 0.59 | 0.60 | 0.60 | **0.61** | **0.61** |
> > > | **OVERALL** | 0.44 | 0.45 | 0.48 | 0.54 | **0.58** |
> > >
> > > To rigorously isolate the source of our model's performance on GeoBench tasks, we conducted an extended ablation study. We compared our base model, the base model augmented with explicit CoT prompting, a domain-agnostic General RL baseline (VLM-R1-7B), which is trained with the original recipe (REC + GUI) in [1] on Qwen2.5-VL-7B, and our Geo-R1 model operating both with and without the thinking prompt. As shown in the table above, general visual reasoning (whether elicited via prompting or post-training) provides only marginal gains. Conversely, Geo-R1 significantly outperforms all baselines even when denied the ability to 'think' explicitly, demonstrating that the robust geospatial priors required for remote sensing tasks have been successfully embedded into the model parameters via our indirect reward mechanism.
> > >
> > > [1] Shen, Haozhan, et al. "Vlm-r1: A stable and generalizable r1-style large vision-language model." arXiv preprint arXiv:2504.07615 (2025).
> > >
> > > >**[Follow-up Question 2]** Regarding Catastrophic Forgetting.
> > >
> > > We appreciate your close attention to the general benchmark results in Appendix L. However, we respectfully disagree with the characterization of this degradation as "catastrophic forgetting."
> > >
> > > In standard continual learning literature, catastrophic forgetting implies an abrupt, severe, and often irrecoverable collapse in previously learned capabilities (e.g., performance dropping near random chance). Our empirical observations in Geo-R1 diverge significantly from this definition.
> > >
> > > While we fully acknowledge the consistent performance declines across general categories (language, math, OCR), these drops are quantitatively modest, typically single-digit percentage decreases. Retaining the vast majority of the base model's general capabilities while simultaneously achieving massive relative accuracy breakthroughs on complex geospatial tasks represents a highly favorable outcome rather than a "catastrophic" failure.
> > >
> > > Instead of a catastrophic collapse, we contextualize this as a necessary "Specialization Trade-off" driven by "Capacity Reallocation." In LLM/VLM post-training, injecting deep, domain-specific causal logic (like geographic perspective projection and cross-view scale mapping) inevitably requires reallocating a small fraction of the model's finite parameter capacity away from encyclopedic general knowledge.
> > >
> > > We agree that transparency regarding this performance shift is critical for the community. Therefore, rather than simply claiming we "avoided" forgetting, we will revise the main text and limitations section to explicitly quantify this Specialization Trade-off. We will clearly state that while Geo-R1 secures state-of-the-art geospatial reasoning, it incurs a minor but measurable cost on general reasoning benchmarks. We believe this framing is empirically accurate and far more constructive for future research than the "catastrophic" label.

---

### Official Review · Reviewer_wFD1 · 2026-03-13

**Soundness:** 2
**Presentation:** 2
**Significance:** 2
**Originality:** 2
**Overall Recommendation:** 3
**Confidence:** 3

**Summary:**

This paper studies whether broad geospatial reasoning can be improved using only an indirect reward. The method first teaches a base vision language model a simple geospatial reasoning style using a small synthetic supervised set built from cross view image pairs. Then it applies reinforcement learning on a panorama to satellite matching task, where the reward is based on whether the final match is correct. The paper argues that solving this proxy task forces the model to learn useful view invariant geospatial cues, and this then transfers to many unseen tasks. Experiments show very strong gains on the matching task itself and large zero shot improvements on several geospatial benchmarks, including street view geolocation, aerial understanding, and some reasoning style tasks.

**Compliance With Llm Reviewing Policy:**

Affirmed.

**Final Justification:**

Score 2 -> 3 after rebuttal.

**Key Questions For Authors:**

Please see weaknesses section.

**Limitations:**

Please see weaknesses section.

**Strengths And Weaknesses:**

**Strengths**

- The paper has a clear and interesting high level idea. Using metadata based cross view matching as a scalable reward is a practical direction for a domain where dense labels are limited.

- The empirical gains on several geospatial benchmarks are strong, especially compared with the base open source model. The paper also tests both ground view and aerial view tasks, which makes the transfer story more interesting.

- The paper is easy to follow at a high level. The two stage pipeline and the intended intuition are simple enough that others could build on the idea.

**Weaknesses**

- The main claim about indirect rewards is not cleanly isolated from direct supervision. The final system first receives supervised synthetic reasoning traces with city and coordinate style answers, and the full model is stronger than the RL only version on important downstream tasks. Because of this, it is hard to tell how much of the final gain truly comes from the indirect reward itself, instead of from the earlier teacher generated geolocation supervision.

- The novelty is somewhat narrow, since most ingredients are existing methods and data sources. In that setting, the paper needs stronger comparisons to simpler alternatives that use the same information. For example, it would be important to compare against straightforward supervised training or other simpler learning setups on the same cross view pairing signal. Without this, it is hard to judge whether the main contribution is a real method advance or mainly a strong application of existing post training tools to a fun domain.

- The zero shot and out of distribution claims still feel under supported. The training data already covers multiple cities and countries, but the paper does not provide a clean analysis of geographic overlap and possible leakage between training priors and evaluation settings. So the results do show transfer to new tasks, but they do not fully prove that the model learned a geography agnostic reasoning rule rather than a mix of useful priors and task transfer.

- I also found the claim about preserving primitive abilities too strong relative to the appendix tables. The extra benchmarks look mixed and in several cases the final model is lower than the base model, so this part of the conclusion should be stated more carefully.

---

> ### Author Rebuttal · Authors · 2026-03-29
>
> We sincerely appreciate the constructive comments from reviewer wFD1 and the time spent on reviewing this paper. We address the questions and clarify the issues accordingly as described below.
>
> >**[W1]** The main claim about indirect rewards is not cleanly isolated from direct supervision...
>
> **[Re W1]**: For the Geo-R1 framework, SFT and RL are both core components. They have distinct roles: SFT primarily provides the model with templates for thinking patterns, while RL, through indirect rewards, primarily provides powerful, generalizable reasoning capabilities. SFT primarily provides a thinking paradigm rather than injecting knowledge about cities, coordinates, and the like. This is because the SFT data consists mainly of synthesized CoT text, while cities and coordinates contribute very little to the total loss. In the GeoChain experiments, we can observe that the improvement of the Geo-SFT model over the base model is very slight.
>
> RL with indirect rewards was the primary driver of the model’s performance improvements. This is evident from the comparison between Geo-R1 and Geo-SFT, as well as the comparison between Geo-R1-Zero and the base model. We have indeed observed instances where Geo-SFT outperforms Geo-R1-Zero on a small subset of GeoChain sub-tasks (Tasks 7 and 9). This is because Tasks 7 and 9 involve two types of relatively subjective classification. Without the scaffolding provided by the thinking paradigm, Geo-R1-Zero exhibited unstable output formats and uncontrolled growth in the CoT. In contrast, Geo-R1, which has undergone SFT and RL training, avoids this issue and consistently outperforms the Geo-SFT model across all subtasks, indicating RL with indirect reward is mainly attributed to for the final gain.
>
> >**[W2]** The novelty is somewhat narrow...
>
> **[Re W2]**: Just as we chose Applications->Earth Sciences as our primary area for this submission, this paper is application-oriented. However, this does not mean that our methodology lacks novelty and innovation; we have pioneered a new approach to training RL models by **utilizing geospatial metadata and associated multi-view images as indirect rewards**.
>
> We appreciate the reviewers’ suggestion to compare our approach with other fine-tuning methods.
> We have supplemented the results on ID test set and IMAGEO-GSS of training using SFT and DPO with the same crossview pairing data, as shown below. We can see that our method demonstrates better task generalization compared to SFT and DPO.
>
> Method|ID_acc|city_acc|coun_acc|mean_dist|median_dist
> -|-|-|-|-|-
> SFT(pairing)|0.81|0.2179|0.6754|976.84|127.90
> DPO|0.67|0.2171|0.6523|1023.81|139.20
> Geo-R1|0.82|0.3272|0.8146|568.32|69.40
>
>
> >**[W3]** The zero shot and out of distribution claims
>
> **[Re W3]**: First, our zero-shot claim emphasizes **task-level generalization**. Evaluation benchmarks use entirely different task formulations, camera parameters, and sampling altitudes compared to our training set.
>
> Second, there are **strictly zero overlapping image samples** between training and evaluation. While minor city-level distribution overlap exists, the vast majority of test data is strictly OOD: GeoChain (10.96% city overlap), IMAGEO-GSS (7.93%), and IMAGEO-UPC (6.42%).
>
> To prove Geo-R1 learned a geography-agnostic reasoning rule rather than memorizing priors, we isolated IMAGEO-GSS performance into overlapping vs. non-overlapping (strictly unseen) subsets:
>
> | Model | Overall | Overlapping (7.93%) | Non-Overlapping (92.07%) |
> | :--- | :---: | :---: | :---: |
> | Qwen2.5-VL-7B (base)| 27.7% | 33.7% | 27.2% |
> | Geo-R1| 32.7% | 37.1% | **32.3%** |
> | Gain | +5.0% | +3.4% | **+5.1%** |
>
> If performance relied on geographic leakage, accuracy would collapse on unseen data. Instead, Geo-R1 maintains 32.3% accuracy on 5,664 strictly non-overlapping samples. Crucially, the **+5.1% absolute gain on unseen cities** exceeds the gain on seen cities (+3.4%). This emprically refutes the memorization hypothesis, proving true zero-shot generalization. We will add this to the next version.
>
> >**[W4]** claim about preserving primitive abilities.
>
> **[Re W4]**: We appreciate the reviewer's careful reading. We agree that the terms "preserves" and "preservation" used in title of section 5.3 and the text were too absolute given the mixed results in the appendix.
>
> Our true intended claim, as also stated in Remark 10, is that our indirect reward RL framework successfully **avoids catastrophic forgetting**. In standard domain-specific full fine-tuning, general-purpose capabilities often suffer severe degradation (e.g., an MMMU score collapsing from 0.6 to 0.2).
>
> In contrast, the performance fluctuations observed in Geo-R1 on general benchmarks are extremely minor. When weighed against the massive state-of-the-art gains achieved in complex geospatial reasoning, this marginal trade-off represents a highly favorable outcome. We will revise the next release to replace the term "preserve" with "mitigating catastrophic forgetting".

---

> > ### Author Rebuttal · Reviewer_wFD1 · 2026-04-03
> >
> > I thank the authors for the detailed rebuttal. My concerns are partially resolved, but the concern of novalty still remains. I will raise my score to 3.

---

> > > ### Author Response · Authors · 2026-04-08
> > >
> > > We are pleased that our rebuttal has, to some extent, addressed the reviewers’ concerns. We appreciate the reviewers’ acknowledgment of our response in the form of a raised score. We will make the necessary adjustments in a subsequent version to reflect the reviewers’ excellent suggestions.

---

### Decision · Program_Chairs · 2026-04-30

**Decision:**

Accept (regular)

**Comment:**

This paper receives three Weak Reject and one Accept. With the competitiveness of ICML, the obtained score has to results in a rejection decision. Hope the authors find the detailed comments and discussions being helpful for improving the work.